# GTPBP8 plays a role in mitoribosome formation in human mitochondria

Miriam Cipullo[1], Genís Valentín Gesé[2], Shreekara Gopalakrishna [1], Annika Krueger [1], Vivian Lobo[3,4], Maria A. Pirozhkova[5], James Marks[6], Petra Páleníková [7], Dmitrii Shiriaev[1], Yong Liu [1], Jelena Misic[1], Yu Cai[1], Minh Duc Nguyen[1], Abubakar Abdelbagi [1], Xinping Li [8], Michal Minczuk [7], Markus Hafner [6], Daniel Benhalevy [5], Aishe A. Sarshad [3,4], Ilian Atanassov [8], B. Martin Hällberg [2] & Joanna Rorbach [1] ✉

Mitochondrial gene expression relies on mitoribosomes to translate mitochondrial mRNAs. The biogenesis of mitoribosomes is an intricate process involving multiple assembly factors. Among these factors, GTP-binding proteins (GTPBPs) play important roles. In bacterial systems, numerous GTPBPs are required for ribosome subunit maturation, with EngB being a GTPBP involved in the ribosomal large subunit assembly. In this study, we focus on exploring the function of GTPBP8, the human homolog of EngB. We find that ablation of GTPBP8 leads to the inhibition of mitochondrial translation, resulting in significant impairment of oxidative phosphorylation. Structural analysis of mitoribosomes from GTPBP8 knock-out cells shows the accumulation of mitoribosomal large subunit assembly intermediates that are incapable of forming functional monosomes. Furthermore, fPAR-CLIP analysis reveals that GTPBP8 is an RNA-binding protein that interacts specifically with the mitochondrial ribosome large subunit 16 S rRNA. Our study highlights the role of GTPBP8 as a component of the mitochondrial gene expression machinery involved in mitochondrial large subunit maturation.

Mitochondria are vital double-membraned organelles present in most eukaryotic cells, and they play a fundamental role in generating adenosine triphosphate (ATP), the primary energy currency for cellular processes. Although most mitochondrial proteins are nuclear-encoded, mitochondria also possess their own small genome, the mitochondrial DNA (mtDNA). To express proteins encoded in the mtDNA, mitochondria employ a specialized translation machinery including tRNAs and a mito-ribosome. Human mitoribosomes synthesize 13 polypeptides, all crucial components in the oxidative phosphorylation (OxPhos) complexes.

The human mitoribosome comprises a 39 S large subunit (mt-LSU) and a 28 S small subunit (mt-SSU). The mt-LSU consists of a 16 S mt-rRNA, mt-tRNAVal, and 52 mitoribosomal proteins (MRPs), while the mt-SSU consists of a 12 S mt-rRNA and 30 MRPs[1,2]. Despite the evolutionary origin of mitochondria from ancient symbiotic bacteria, mitoribosomes differ significantly from bacterial as well as cytosolic

[1]Department of Medical Biochemistry and Biophysics, Karolinska Institutet, Stockholm 17165, Sweden. [2]Department of Cell and Molecular Biology, Karolinska Institutet, Solnavägen 9, Stockholm 17165, Sweden. [3]Department of Medical Biochemistry and Cell Biology, Institute of Biomedicine, University of Gothenburg, SE-40530 Gothenburg, Sweden. [4]Wallenberg Centre for Molecular and Translational Medicine, University of Gothenburg, SE-40530 Gothenburg, Sweden. [5]Lab for Cellular RNA Biology, Shmunis School of Biomedicine and Cancer Research, George S. Wise Faculty of Life Sciences, Tel Aviv University, Tel Aviv, Israel. [6]Laboratory of Muscle Stem Cells and Gene Regulation, National Institute of Arthritis and Musculoskeletal and Skin Diseases, National Institutes of Health, Bethesda, MD, USA. [7]MRC Mitochondrial Biology Unit, University of Cambridge, Cambridge CB2 0XY, UK. [8]Proteomics Core Facility, Max-Planck-Institute for Biology of Ageing, Joseph-Stelzmann-Str. 9b, 50931 Cologne, Germany. ✉e-mail: joanna.rorbach@ki.se

ribosomes. The presence of mitochondrial-specific proteins, mitochondrial N- and C-terminal extensions of universal ribosomal proteins, the reduced size of mt-rRNAs as well as the different chemical compositions of the peptide exit tunnel and the mRNA entry site, highlight functional and structural divergences. Additionally, the fact that the mtDNA encodes rRNAs and tRNAs, whereas all 82 MRPs are nuclear-encoded suggests the involvement of sophisticated mechanisms for the coordinated synthesis and assembly of this complex machinery[3].

Cryo-electron microscopy (cryo-EM) has recently played a significant role in the structural characterization of mitoribosomes, allowing for the resolution of various stages of the late-stage assembly of the mt-LSU and mt-SSU[4]. An in-depth understanding of these processes is of high importance as several mitochondrial illnesses have been linked to mutations in MRPs and mitoribosome assembly factors, and additional research is needed for this purpose[5].

The biogenesis of mitoribosomes is facilitated by several auxiliary proteins, including rRNA-modifying enzymes such as methyltransferases and pseudouridine synthases, as well as chaperones, helicases, and GTP-binding proteins (GTPBPs). These proteins contribute to the efficient assembly and maturation of the mitoribosome structure.

GTPBPs, a class of regulatory proteins, are widely found in all domains of life. These proteins play crucial roles in various cellular functions, and their catalytic activity is dependent on the hydrolysis of GTP to GDP through a conserved G-domain. The G-domain consists of five motifs (G1-G5), among which the Walker A motif (G1/P-loop) is exceptionally conserved and responsible for nucleotide binding, ensuring the proper functioning of GTPBPs[6]. In recent years, there has been a notable increase in studies investigating the role of GTPBPs in mitoribosome biogenesis within mammalian mitochondria. For example, regarding the mt-SSU, NOA1 (C4orf14) has been identified as a GTP-dependent interactor with the small subunit[7,8]. Together with ERA-like 1 (ERAL1), another GTPase that directly binds to 12 S mt-rRNA[9,10], NOA1 directs the maturation of the mt-SSU platform and decoding center[11]. As for the mt-LSU, GTPBP7 (MTG1) contributes to the maturation of the peptidyl-transferase center (PTC) and prevents the subunit joining[12–14], while GTPBP6 guides the final maturation steps of the PTC[15,16]. Late-state mt-LSU assembly also involves the two Obg-family proteins GTPBP10 and GTPBP5[17–22], with the latter directly contributing to the final remodeling of the PTC[13,16,23].

Another GTPBP identified in mammals with yet unknown function is GTPBP8. Its bacterial homolog belongs to the EngB family of GTPBPs (also known as YihA/YsxC), which play an important role in regulating cell division[24]. Moreover, they have been shown to contribute to the assembly of the 50 S ribosomal subunit in various prokaryotic species[25–28]. These proteins solely consist of the G-domain[29,30] and feature a highly charged C-terminal alpha-helix which is thought to be important in mediating the binding of EngB to the large subunit (Supplementary Fig. 1A)[29,31]. A recent structural analysis of the late stages of assembly of the mitochondrial large subunit in *Trypanosoma brucei* showed an EngB homolog interacting with the subunit interface via its long C-terminal extension (CTE) (Supplementary Fig. 1A)[32]. In *S. cerevisiae*, however, the homolog of EngB, Mrx8, is important for translating COX1 transcript in cold-stress conditions[33].

In this study, we aim to elucidate the function of the human homolog of EngB, GTPBP8. We find that it localizes in mitochondria where it interacts with several proteins involved in mitochondrial gene expression. Additionally, we perform knock-out experiments of GTPBP8 in human cells, which results in pronounced mitochondrial dysfunction, as evidenced by a significant defect in the OxPhos system. Through proteomic and structural analyses of GTPBP8 knock-out cells, we identify a disruption in the mitoribosome assembly pathway, leading to a considerable decrease in monosome formation and subsequently reduced mitochondrial translation. Furthermore, the fluorescent-based protocol of Photo-Activatable Ribonucleotide Cross Linking and Immuno-Precipitation (fPAR-CLIP), reveals a highly specific interaction with the 16 S mt-rRNA of the mt-LSU. Overall, our data demonstrate that GTPBP8 is essential for mitochondrial viability. Acting as an RNA-binding protein, it exhibits specific interactions with the mt-LSU 16 S mt-rRNA and its absence results in impairments in mt-LSU assembly.

## Results

### GTPBP8 localizes in the mitochondrial matrix and interacts with the mitochondrial gene expression machinery

Previous immunoprecipitation analysis using mitochondrial RNA granules (MRGs) component DDX28 as bait has identified GTPBP8 to colocalize with these foci in human cells[34]. Moreover, the yeast orthologue of GTPBP8, Mrx8, was shown to localize in mitochondria[33]. To confirm GTPBP8 mitochondrial subcellular localization (as it has not yet been annotated in the human MitoCarta3.0 dataset), we performed immunofluorescence imaging of human 143B osteosarcoma (HOS) cells transiently expressing a C-terminal FLAG-tagged *GTPBP8* cDNA construct (*GTPBP8::FLAG*) (Supplementary Fig. 1B). GTPBP8::FLAG co-localized with TOM20, an outer mitochondrial membrane protein, suggesting mitochondrial localization of GTPBP8 (Supplementary Fig. 1B). Based on this observation, we next generated Flp-In T-REx HEK293 cells stably transfected with C-terminal FLAG-tagged *GTPBP8* and performed subcellular fractionation. The FLAG signal was greatly enriched in the mitochondrial fraction, in contrast to the cytosolic ribosomal protein uL10 (Supplementary Fig. 1C). Moreover, upon Proteinase K treatment on isolated mitochondria, the FLAG signal remained intact, following the same pattern as mitochondrial matrix proteins (mL37, uS16m), whereas uL10 and TOM20 were both digested by Proteinase K (Supplementary Fig. 1C). This strongly indicates the localization of GTPBP8 within mitochondria in human cells. As our sequence homology analysis (Supplementary Fig. 1D), confirmed that GTPBP8 is related to bacterial EngB GTPases, we therefore next sought to determine whether GTPBP8 has a similar role in mammalian mitochondrial gene expression.

To identify GTPBP8 binding partners, we performed a FLAG-immunoprecipitation (FLAG-IP) experiment followed by label-free quantitative (LFQ) mass spectrometry analysis of HEK293 cells overexpressing GTPBP8::FLAG. C-terminally FLAG-tagged mitochondrially targeted luciferase (mtLuc::FLAG) was used as a control (Fig. 1A). A similar setup allowed us to previously identify GTPBP10 and GTPBP5 as the components of the mt-LSU assembly pathway[19,21]. In contrast to GTPBP10 and GTPBP5 results, we did not observe significant enrichment of mitoribosomal proteins in the GTPBP8::FLAG pull-downs, except for uS15m and uL16m (Fig. 1A, Supplementary Data 1). Interestingly, several proteins involved in RNA metabolism and post-transcriptional modifications (TRMT10C, MTPAP, MTERF1, RPSUD4), as well as mitoribosome assembly (GTPBP10, MRM1, DDX28, MRM3, DHX30, MTERF3, TFB1M, NOA1, METTL17, ERAL1), were enriched by > 8 fold (Fig. 1A), with statistically significant enrichment of mitoribosome biosynthesis related pathways confirmed via Gene Set Enrichment Analysis (GSEA) (Fig. 1B). Of note, overexpression of GTPBP8::FLAG protein did not affect MRPs steady-state levels or monosome formation (Supplementary Fig. 1E and S1F).

GTPBPs have been reported to bind to the ribosome in a GTP-dependent manner. Since the hydrolysis of GTP can occur in the milliseconds range[35,36], the majority of these interactions are considered transient. Therefore, to further assess GTPBP8 interactome network, we complemented our FLAG-IP analysis with a proximity labeling (BioID) assay, previously employed for the detection of fast-occurring interactions[37]. We generated Flp-In T-REx HEK293 cell lines stably transfected with GTPBP8 fused to C-terminally HA-tagged BirA* protein (GTPBP8-BirA*). We also engineered the GTPBP8^{S124A}-BirA* cell line, with a mutation of a highly conserved residue (S124A) of the

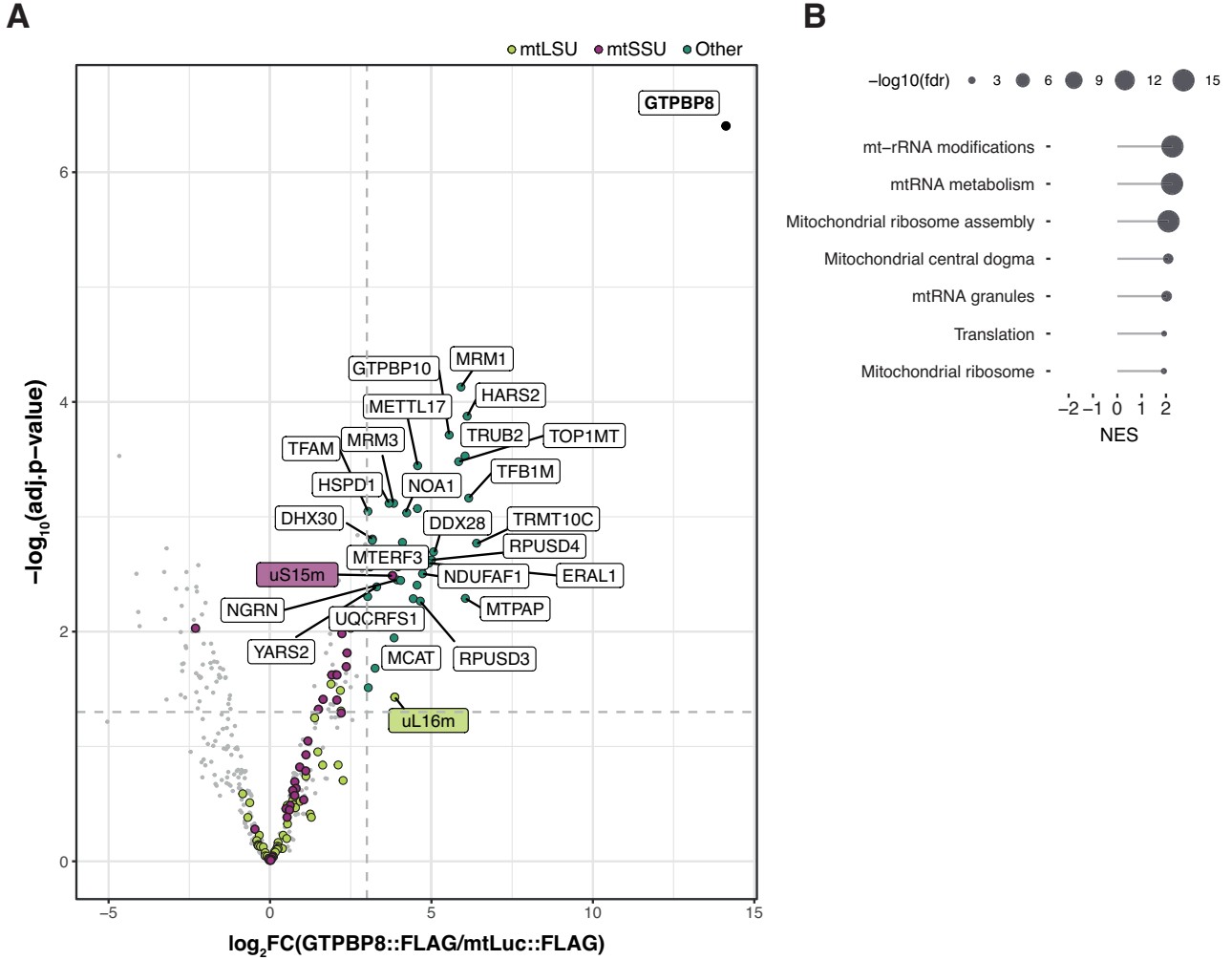

**Fig. 1 | GTPBP8 interacts with proteins involved in mt-rRNA metabolism and mitoribosome assembly. A** Label-free quantitative (LFQ) mass spectrometry analysis following FLAG-IP of GTPBP8::FLAG ($n = 3$ independent experiments). Eluates from mtLuc::FLAG (control) and GTPBP8::FLAG were used. The $\log_2$(fold change) and the $-\log_{10}$(adjusted $p$-value) are displayed on the $x$- and $y$-axis, respectively. MRPs from the mt-LSU (light green) and the mt-SSU (purple) are highlighted as well as other proteins above the threshold (turquoise). The dashed lines (light gray) represent the thresholds of the $\log_2 FC > 3$ and $-\log_{10}$(adj.$p$-value) > 1.3 (adj.$p$-value < 0.05). The volcano plot was filtered for mitochondrial proteins as annotated in MitoCarta 3.0. A full annotation of the identified proteins is displayed in Supplementary Data 1. **B** GSEA analysis of the FLAG-IP results was performed using the Webgestalt R package. The enrichment score normalized (NES) is indicated together with the $-\log_{10}$ of the false discovery rate (FDR).

Walker A motif of GTPBP8's G-domain, responsible for GTP-hydrolysis, with the intention of better capturing the interactions. As a control, we used BirA* targeted to mitochondria (MTS-BirA*) (Supplementary Fig. 2A and S2C, Supplementary Data 2). With this approach, as expected, we enriched for a higher number of mitochondrial hits overall. Interestingly, more MRPs were enriched, predominantly from the mt-LSU, in both the GTPBP8-BirA* and the GTPBP8$^{S124A}$-BirA* cell lines (Supplementary Fig. 2A and S2C). Most of the proteins detected with adjusted $p$-value < 0.05 and $\log_2 FC > 3$ are components of the MRGs and are involved in different pathways of RNA metabolism and mitoribosome assembly, confirming our previous data from FLAG-IP experiments (Fig. 1 and Supplementary Fig. 2). In addition, gene enrichment analysis using the GSEA method also revealed the presence of proteins involved in mitochondrial translation and OxPhos assembly, showing no major differences between GTPBP8-BirA* and GTPBP8$^{S124A}$-BirA* (Supplementary Fig. 2B and S2D).

**Loss of GTPBP8 causes a severe OxPhos defect**

Concurrently with our interactome analysis, we employed CRISPR/Cas9 technology to create knock-out cell lines of GTPBP8

(Supplementary Fig. 3A). This was undertaken to explore the impact of GTPBP8 loss on mitochondrial function. The first protein-coding exon of the *GTPBP8* gene, which encodes for the GTPBP8 protein, was used as a target for two sgRNAs designed to cause an out-of-frame deletion (Supplementary Fig. 3A). Two knock-out clones (GTPBP8$^{KO1}$ and GTPBP8$^{KO2}$) were confirmed (Fig. 2A and Supplementary Fig. 3) and used for further analyses.

First, we investigated the proliferation rates of GTPBP8 knock-out cells in media supplemented with galactose instead of glucose, which forces cells to rely mainly on mitochondrial respiration for ATP synthesis (Fig. 2B). In comparison to the wild-type HEK293 cell line, GTPBP8$^{KO1}$ and GTPBP8$^{KO2}$ capability to grow in galactose media was strongly diminished, suggesting the ablation of GTPBP8 causes an OxPhos defect (Fig. 2B). To further corroborate the OxPhos deficiency, we performed western blotting from cell lysates of HEK293, GTPBP8$^{KO1}$ and GTPBP8$^{KO2}$ cell lines (Fig. 2C). The steady-state levels of Complex IV protein MTCO2 (mtDNA-encoded) and Complex I protein NDUFB8 (nuclear-encoded) were strongly reduced in GTPBP8 knock-out cell lines compared to control, whereas Complex V protein ATP5A (nuclear-encoded) levels were unchanged (Fig. 2C). Additionally, the

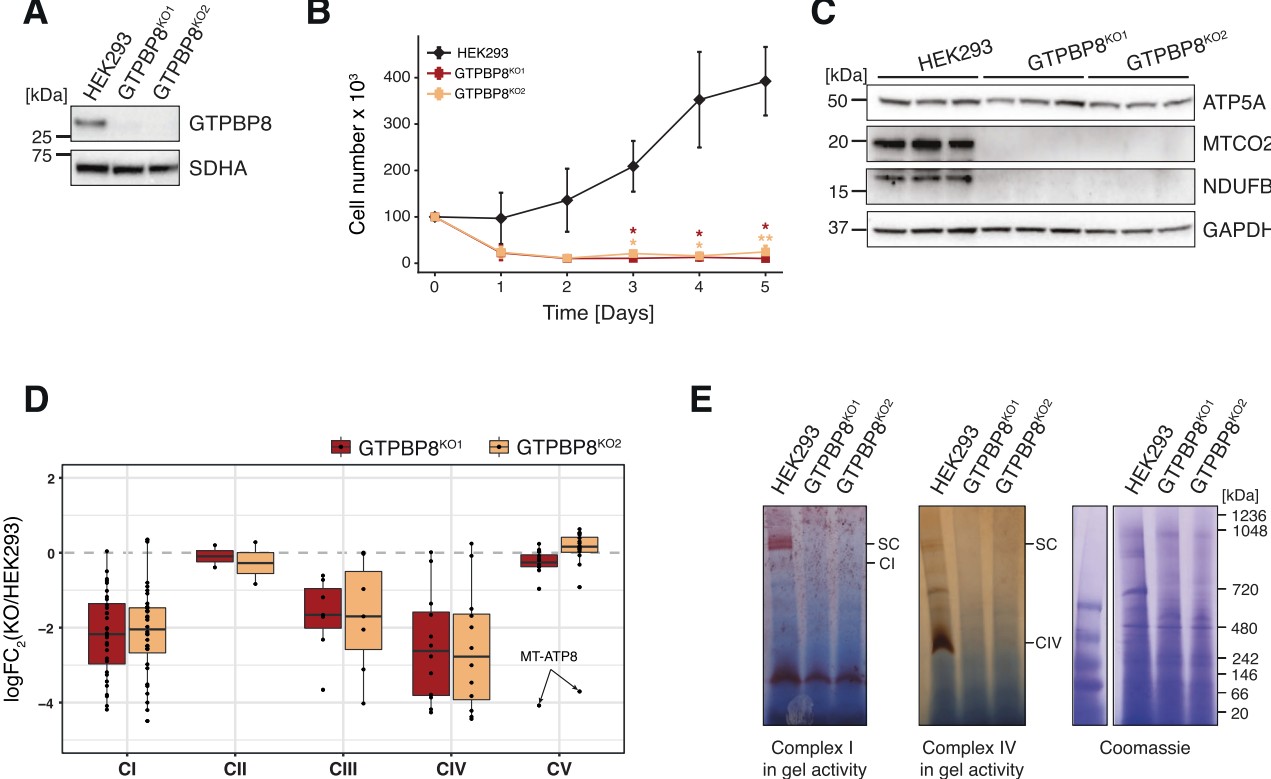

**Fig. 2 | Ablation of GTPBP8 causes an OxPhos deficiency. A** Western blotting to confirm GTPBP8 knock-out. Whole-cell lysates from HEK293, GTPBP8[KO1] and GTPBP8[KO2] were used for SDS-PAGE analysis. GTPBP8 antibody was used to detect GTPBP8 endogenous levels; SDHA antibody was used as a loading control. **B** Growth rates of GTPBP8 knock-out cells in galactose medium. HEK293 (black), GTPBP8[KO1] (red), and GTPBP8[KO2] (yellow) were cultured in DMEM containing 0.9 g/l galactose ($n = 3$ independent experiments). The mean average cell number per each time point is indicated, and error bars = ± 1 standard deviation (SD). Statistical analyses were performed using the Student's two-tailed $t$-test. $P$-values and source data are provided in the Source Data file. **C** Western blotting to assess OxPhos complexes steady-state levels in GTPBP8 knock-out cell lines. Whole-cell lysates from HEK293, GTPBP8[KO1] and GTPBP8[KO2] were resolved via SDS-PAGE. Membranes were probed using the OxPhos antibody cocktail against nuclear-encoded proteins of complexes V (ATP5A) and I (NDUFB8) and mtDNA-encoded complex IV protein MTCO2. GAPDH was used as a loading control. **D** LFQ mass spectrometry analyses of OxPhos complexes in GTPBP8 knock-out cell lines compared to HEK293 ($n = 3$ independent experiments). Log$_2$(FC) of GTPBP8[KO1] vs HEK293 (red) and GTPBP8[KO2] vs HEK293 (yellow) is represented on the $y$-axis for each OxPhos complex. Each dot refers to a protein, which constitutes the specific complex. A summary of boxplots is provided in a Source Data file. All the log$_2$(FC) values are listed in Supplementary Data 3. **E** Assessment of OxPhos complex I and complex IV activities in GTPBP8 knock-out cell lines compared to control. Mitochondrial lysates from HEK293, GTPBP8[KO1] and GTPBP8[KO2] cell lines were resolved by BN-PAGE and stained using complex I (CI) and complex IV (CIV) -specific substrates ($n = 1$ independent experiment). Coomassie staining was used as a loading control. SC supercomplexes. Source data are provided as a Source Data file.

label-free quantification (LFQ) proteomic analysis of GTPBP8[KO1] and GTPBP8[KO2] mitolysates in comparison with HEK293 control mitolysates confirmed an overall strong reduction of Complex I, Complex III, and Complex IV in the absence of GTPBP8, while Complex II and Complex V were stable (Fig. 2D, Supplementary Data 3). Of note, MT-ATP8, the only detected component of Complex V that is mitochondrially-encoded, was significantly reduced, further indicating a specific problem with the expression of the mitochondrial genome (Fig. 2D). The in-gel activity of OxPhos complexes also showed a reduction in Complex I and Complex IV activity (Fig. 2E), providing further evidence that GTPBP8 ablation causes a severe OxPhos impairment.

### GTPBP8 is important for mitochondrial translation

Given that GTPBP8 knock-out cell lines display a clear OxPhos alteration, we next determined whether this phenotype was a consequence of decreased rates of mitochondrial translation. GTPBP8[KO1] and GTPBP8[KO2] were found to have a strong reduction of synthesis of mtDNA-encoded proteins compared to the control, as assessed by [35S]-metabolic labeling of mitochondrial translation products (Fig. 3A).

To obtain further insights into the translation defect, we performed Mitochondrial Ribosome Profiling (MitoRiboSeq) analysis in GTPBP8[KO1] and HEK293. Through this approach, we detected a highly

decreased incidence of mitoribosome-protected fragments (mt-RPFs) on all mitochondrial transcripts (Fig. 3B), in agreement with the [35S] translation assay (Fig. 3A). For most of the transcripts, the levels of mt-RPFs in GTPBP8[KO1] were below 25% compared to control levels, apart from the ATP8 and ND4 open reading frames (ORFs). Closer inspection of these transcripts revealed increased levels of RPFs at the start codons of ATP6 and ND4 ORFs in comparison to control cells (Supplementary Fig. 3B), contributing to the higher total number of mt-RPFs on ATP8/ATP6 and ND4L/ND4 transcripts in comparison to other transcripts. Together, these results suggest that there are very few mitoribosomes associated with the transcripts, and even though some initiation complexes are formed, mitoribosomes are not fully functional as there was an impaired transition from the initiation to the elongation step of translation (Supplementary Fig. 3B). Of note, due to the lack of 5' UTRs in most mitochondrial transcripts, the occupancy of mitoribosomes on start codons of other ORFs could not be analyzed.

To ascertain whether a transcriptome defect underlies the observed decrease in protein synthesis, the steady-state levels of mt-mRNAs and mt-tRNAs were examined by quantitative real-time PCR and northern blotting respectively (Fig. 3C and Supplementary Fig. 3C). Interestingly, the steady-state levels of several mt-mRNAs were altered, with some being significantly upregulated (ND1) and

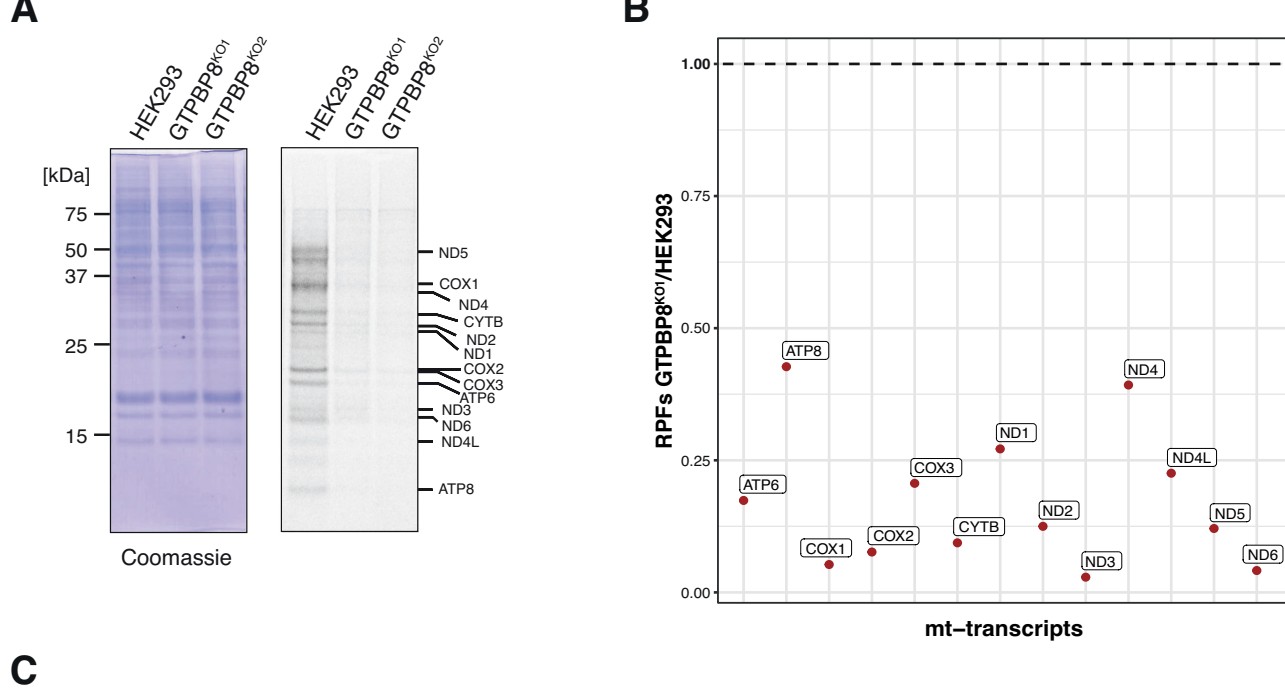

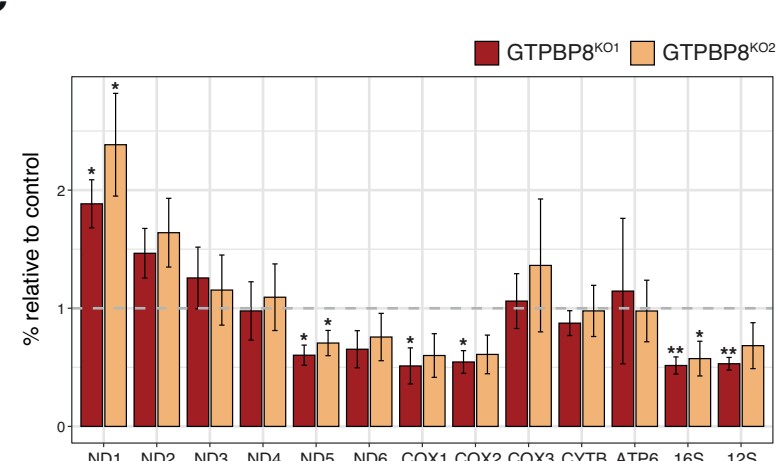

**Fig. 3 | Mitochondrial protein synthesis is decreased in the absence of GTPBP8.**
**A** In vivo translation assay of GTPBP8$^{KO1}$ and GTPBP8$^{KO2}$ cell lines compared to HEK293. Cells were labeled with a [$^{35}$S]-methionine and cysteine mix upon cytosolic translation inhibition. Whole cell lysates were resolved by SDS-PAGE and the signal was later detected via autoradiography. Loading was determined using Coomassie staining ($n$ = 3 independent experiments). **B** Mitoribosome profiling analysis of GTPBP8$^{KO1}$ compared to HEK293. The ratio of the mitoribosome-protected fragments (RPFs) in GTPBP8$^{KO1}$ vs control for each mitochondrial transcript is represented on the $y$-axis. The data were determined via MitoRiboSeq and refer to a single experiment. **C** Quantitative real-time PCR of mt-mRNAs and mt-rRNAs steady-state levels in HEK293 cells depleted of GTPBP8 compared to control. Total RNA isolated from HEK293, GTPBP8$^{KO1}$ and GTPBP8$^{KO2}$ was used ($n$ = 3 biological replicates, data presented as mean values ± 1 SD). Student's two-tailed $t$-test was performed. $P$-values and source data are provided in a Source Data file. The significance cut-off was set at $p < 0.05$, with *$p < 0.05$, **$p < 0.01$, and ***$p < 0.001$. Source data are provided as a Source Data file.

some downregulated (ND5, COX1, COX2) (Fig. 3C). The increase in ND1 and ND2 steady-state levels is often present in different models of mitochondrial gene expression defects[10,34,38], and the same has been observed for the downregulation of ND5, COX1, and COX2[10]. However, considering that the translation assay and ribosome profiling show a reduction of translation of all mtDNA-encoded transcripts (Fig. 3A, B), these changes in transcript levels can represent secondary/compensatory effects. Furthermore, this might be influenced by the different turnover rates of specific mt-mRNAs, which are known to be dependent on their translation[39]. In contrast, mt-tRNA steady-state levels were unchanged in both GTPBP8$^{KO1}$ and GTPBP8$^{KO2}$ cell lines compared to HEK293 controls, indicating that the depletion of GTPBP8 does not affect tRNA processing and turnover (Supplementary Fig. 3C). Notably, the deletion of *GTPBP8* caused a substantial reduction in the 12 S and

16 S mt-rRNA steady-state transcript levels (Fig. 3C and Supplementary Fig. 3D).

### GTPBP8 ablation affects mitoribosome formation

As reduced mt-rRNAs steady-state levels may underly a mitoribosome defect, we investigated MRP steady-state levels from both the mt-LSU and mt-SSU. Indeed, western blotting revealed that the steady-state levels of most of the MRPs analyzed were decreased in both GTPBP8 knock-outs (Fig. 4A and 4B), in agreement with the decrease in mt-rRNAs levels (Fig. 3C and Supplementary Fig. 3D).

To obtain a more comprehensive picture, we performed LFQ proteomic analysis on GTPBP8$^{KO1}$ and GTPBP8$^{KO2}$ mitochondrial lysates compared to HEK293 control. This further confirmed a reduction of the majority of MRPs from both the mt-SSU and mt-LSU

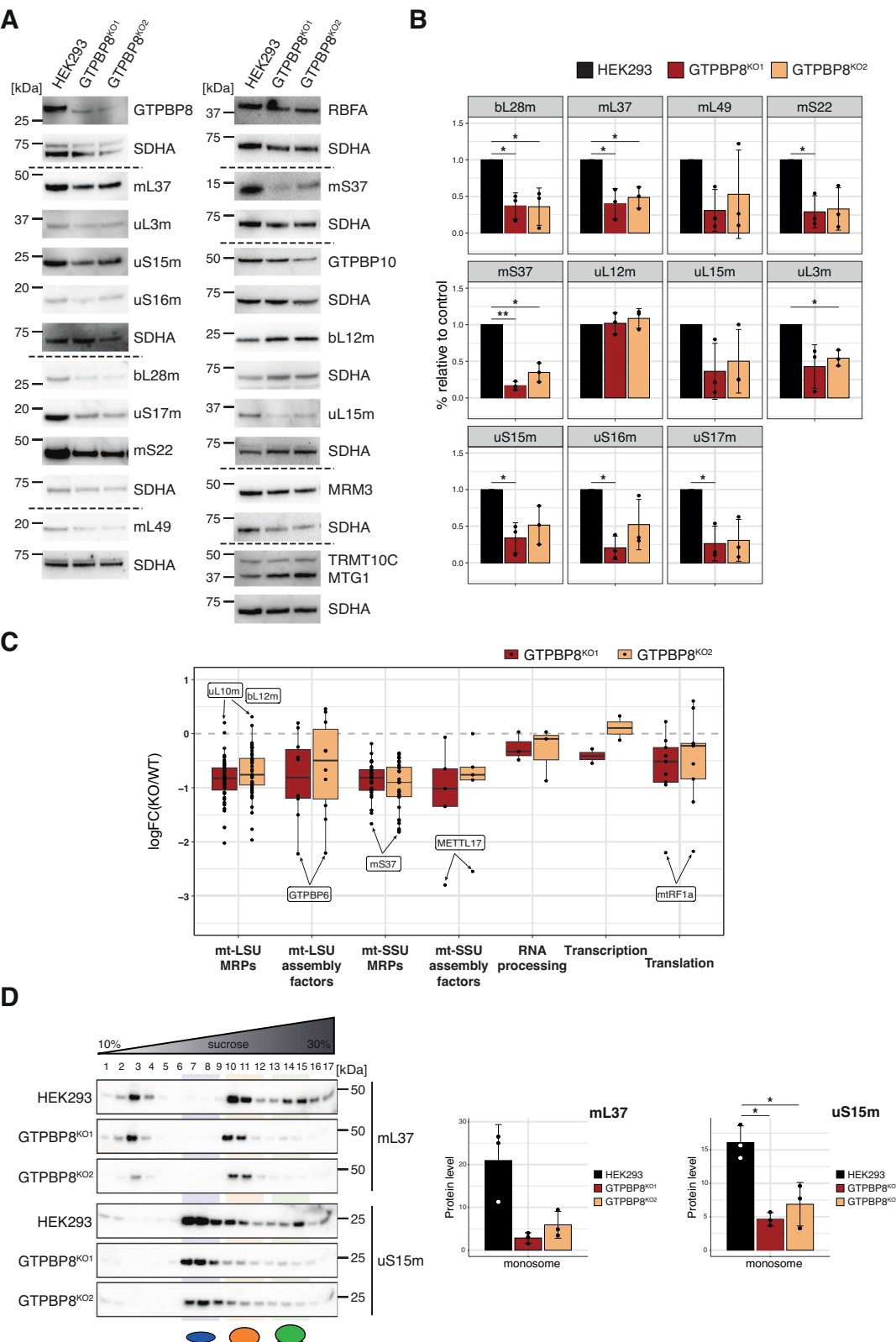

(apart from uL10m and uL12m, which are part of the L7/L12 stalk of the mt-LSU) (Fig. 4C, Supplementary Data 3). Interestingly, mS37 was found to be the most downregulated MRP of the mt-SSU, as shown in previous models of disrupted mt-LSU assembly[40,41]. Since some MRPs were only slightly downregulated, we plotted MRPs steady-state levels based on their recruitment during the assembly of mt-LSU and mt-SSU, as reported by Bogenhagen et al.[3]. The MRPs that are

recruited at the late-stage mt-LSU and mt-SSU assembly were, in general, more noticeably downregulated, while early mt-LSU and mt-SSU MRPs were more stable (Supplementary Fig. 4A), suggesting accumulation of the earlier assembly intermediates in the absence of GTPBP8.

The steady-state levels of mt-LSU and mt-SSU assembly factors exhibited varying changes, as determined by western blotting and

**Fig. 4 | 55 S monosome formation is compromised in the absence of GTPBP8.**
**A** Western blotting analyses of the mt-LSU and mt-SSU MRPs and selected assembly factors steady-state levels in GTPBP8-depleted cells. Whole-cell lysates from HEK293, GTPBP8[KO1] and GTPBP8[KO2] were resolved via SDS-PAGE and membranes were blotted with antibodies against GTPBP8, mt-LSU MRPs (mL37, uL3m, bL28m, mL49, uL12m, uL15m), mt-SSU MRPs (uS15m, uS16m, uS17m, mS22, mS37) and others (RBFA, GTPBP10, MRM3, TRMT10C, MTG1). SDHA was used as a loading control. **B** Quantification of MRPs steady-state levels in GTPBP8 knock-out cells compared to control HEK293. Quantification from western blotting ($n = 3$ biological replicates) was performed using the Image Lab software and protein signals were normalized to SDHA. Student's two-tailed $t$-test was performed. Source data and $P$-values are provided in a Source Data file. **C** LFQ mass spectrometry analyses of MitoCarta3.0 mito-pathways in GTPBP8 knock-out cell lines compared to HEK293 ($n = 3$ independent experiments). Log$_2$(FC) of GTPBP8[KO1] vs HEK293 (red) and GTPBP8[KO2] vs HEK293 (yellow) is represented on the $y$-axis. Each dot refers to a

protein belonging to mt-LSU MRPs, mt-LSU assembly factors, mt-SSU MRPs, mt-SSU assembly factors, mt-RNA processing proteins, mt-transcription and mt-translation pathways. A summary of boxplots is provided in a Supplementary Data File. The complete list of log$_2$(FC) values is presented in Supplementary Data 3. **D** Sucrose gradient centrifugation analysis to assess mitoribosome sedimentation patterns in GTPBP8[KO1] and GTPBP8[KO2] compared to HEK293 control cells. Mitochondrial lysates were loaded onto 10–30 % isokinetic sucrose gradients and obtained fractions were analyzed via western blotting. Membranes were probed for mt-LSU MRP mL37 and mt-SSU MRP uS15m. Below, quantification using Image Lab software of mL37 and uS15m in the monosome fractions from independent replicates is represented ($n = 3$ independent experiments). The samples derived from the same experiment were processed in parallel. The $y$-axis represents the signal detected in the monosome fraction normalized for the total signal of each sample's gradient. Data are presented as mean values +/− SD. Student's two-tailed $t$-test was performed. $P$-values and source data are provided as a Source Data file.

proteomic analyses (Fig. 4A and 4C). Among the mt-LSU assembly factors, GTPBP10, MRM3, DHX30, and DDX28 levels remained largely unchanged, while GTPBP6 showed a significant decrease in GTPBP8 knock-out cell lines (Fig. 4A and 4C). The mt-SSU assembly factors displayed a general reduction, with METTL17 being the most significantly affected protein (Fig. 4C). The expression of proteins involved in mt-RNA processing and transcription remained stable, while some translation factors exhibited slight downregulation (except for a pronounced decrease in mtRF1a; Fig. 4C).

To further investigate how the altered MRPs and mt-rRNA steady-state levels affect mitoribosome formation, we performed sucrose gradient centrifugation experiments on mitochondrial lysates from HEK293, GTPBP8[KO1] and GTPBP8[KO2] (Fig. 4D). GTPBP8[KO1] and GTPBP8[KO2] cells displayed a strong reduction in 55 S monosome formation compared to control cells, as observed from three independent experiments (Fig. 4D). As expected, the steady-state levels of the tested mt-LSU and mt-SSU MRPs (mL37 and uS15m, respectively) were lower. Interestingly, they largely accumulated in the fractions corresponding to the individual subunits/ assembly intermediates, hence, in the absence of GTPBP8, monosome assembly is hindered.

## GTPase activity is essential for GTPBP8 function

In order to support the specificity of the observed mitochondrial defects to GTPBP8 ablation, we generated a GTPBP8[KO1] cell line re-expressing the GTPBP8::FLAG construct (GTPBP8[RESCUE_wt]) via the Flp-In T-REx system (Fig. 5A). In parallel, to further study the effect of S124A mutation in the GTPBP8's G-domain, we re-expressed GTPBP8[S124A]::FLAG construct in the GTPBP8[KO1] cell line to test whether the mutation affects the function of GTPBP8 (GTPBP8[RESCUE_S124A]) (Fig. 5A). Steady-state levels of MTCO2 and NDUFB8 were fully restored upon expression of wild-type (WT) GTPBP8 after induction with both 1 ng/ml and 50 ng/ml doxycycline when compared to the HEK293 control cell line (Fig. 5B). However, the GTPBP8[RESCUE_S124A] cell line was not able to rescue the phenotype under the same conditions, suggesting that the mutation impairs GTPBP8 functionality (Fig. 5B). To corroborate these data, we performed sucrose density ultracentrifugation experiments on HEK293, GTPBP8[KO1], GTPBP8[KO2], GTPBP8[RESCUE_wt] and GTPBP8[RESCUE_S124A] cell lines, and observed that GTPBP8[RESCUE_wt] partially restores monosome formation, compared to control, whereas GTPBP8[RESCUE_S124A] displays a similar defect as compared to the knock-out cell lines (Fig. 5C). Analogous results were obtained from [35S]-labeling assay; the mitochondrial translation defect was restored in GTPBP8[RESCUE_wt], whilst it was comparable to GTPBP8[KO1] and GTPBP8[KO2] in GTPBP8[RESCUE_S124A] (Fig. 5D). In conclusion, the mitochondrial dysfunction examined in GTPBP8 depleted cells is specific to GTPBP8 and the mutation in GTPBP8 G-domain impairs GTPBP8 functionality.

## Mitoribosome assembly intermediates accumulate in GTPBP8 knock-out cells

As our biochemical analyses strongly suggest that GTPBP8 is implicated in the maturation of the mitoribosomal subunits, we next set to determine the structural composition of mt-LSU and mt-SSU assembly intermediates accumulated in the GTPBP8[KO1] cell line. We took advantage of our previously optimized protocols[13] and purified mt-LSU assembly intermediates by performing sucrose gradient ultracentrifugation and subjected the mt-LSU fractions to single-particle cryo-EM (Fig. 6A and Supplementary Fig. 5 and S6). Due to the strong decrease in the levels of MRPs of mt-SSU, this method was not efficient in purifying enough mt-SSU particles for structural studies. Therefore, we performed FLAG-IP of C-terminally FLAG-tagged mS27 (mS27::FLAG) overexpressed in the GTPBP8[KO1] cell line (Supplementary Fig. 7A) and subjected the enriched mt-SSU complexes for cryo-EM (Fig. 6B and Supplementary Fig. 7B). Although the yield of the mt-SSU particles in GTPBP8[KO1] expressing mS27::FLAG was strongly reduced compared to HEK293 cell overexpressing mS27::FLAG (Supplementary Fig. 7A), we were able to collect enough particles for processing.

For the mt-LSU, we identified two distinct classes of mt-LSU sub-assemblies, both displaying a disordered intersubunit interface as compared to the highly ordered solvent side (Fig. 6A). Based on the absence of specific 16 S mt-rRNA helices we were able to classify the mt-LSU populations into State 1 and State 2 (Fig. 6A). The identified states lack bL36m, which is recruited as the last MRP during mt-LSU assembly, as well as rRNA helices H33-35a (1917–2001 nt), H67-71 (2539–2649 nt), and H89-93 (2935–3099 nt), which constitute the enzymatic core of the mt-LSU, namely the peptidyl-transferase center (PTC), resembling the earliest mt-LSU intermediates reported thus far[20,42,43]. In addition, State 1 lacks helices H80-81 (2810–2849 nt), H65 (2475–2506 nt), and 1709–1736 nt in domain I, which are present in State 2 (Fig. 6A and Supplementary Fig. 5). The structure of the most immature mt-LSU assembly intermediate determined to date exhibits disordered 16 S mt-rRNA domains IV and V[20] as observed in State 1, suggesting an accumulation of very early mt-LSU assembly intermediates in our model (Fig. 6A).

As for the mt-SSU, we were able to classify the particles into four different classes which represent a mixture of late mt-SSU assembly intermediates and mature mt-SSU (Fig. 6B). Specifically, 37 % of the total amount of particles represent mt-SSU sub-assemblies, all of which lack the last MRP recruited to mt-SSU, mS37 (State 1 and State 2). Of the 37 %, 6 % displayed density for the known mt-SSU assembly factor METTL15 (State 2) (Fig. 6B), supporting previous findings that showed METTL15 recruitment during the last steps of the assembly process[40]. Interestingly, more than half of the mt-SSU dataset represents fully mature mt-SSU, with 42 % of the particles resembling the previously described mitochondrial pre-

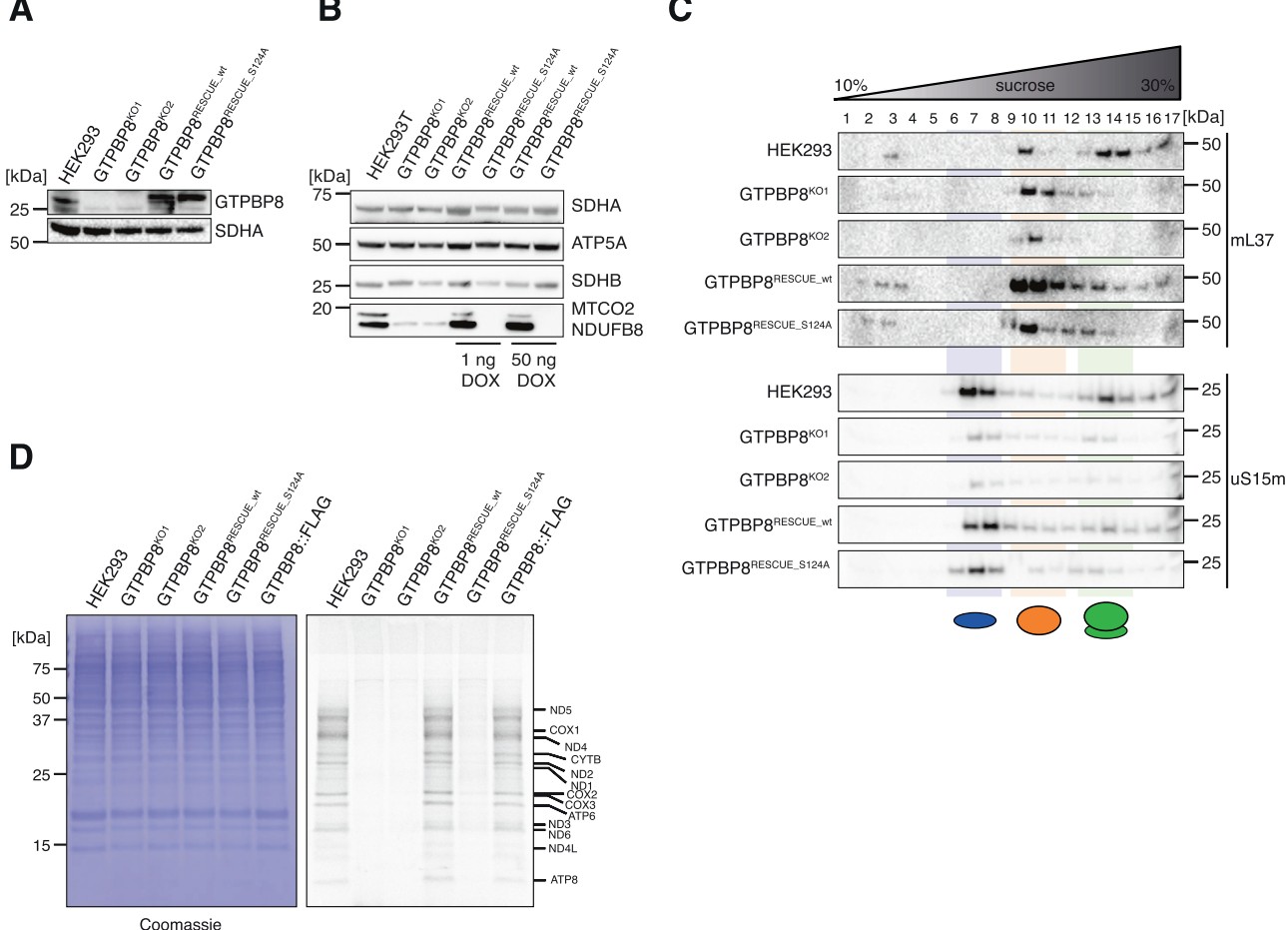

**Fig. 5 | GTP hydrolysis is important for GTPBP8 function. A** Western blotting analysis of control HEK293, GTPBP8[KO1], GTPBP8[KO2], GTPBP8[RESCUE_wt] and GTPBP8[RESCUE_S124A] samples to confirm GTPBP8 knock-out compared to the rescue cell lines. Anti-GTPBP8 antibody was used to compare GTPBP8 endogenous expression with the overexpression in rescue cell lines. SDHA was used as a loading control ($n = 1$ independent experiments). **B** Western blotting analysis to test OxPhos steady-state levels in GTPBP8[KO1] and GTPBP8[KO2] compared to wild-type HEK293 and rescue cell lines GTPBP8[RESCUE_wt] and GTPBP8[RESCUE_S124A]. Overexpression was induced prior to the experiment using 1 ng/ml and 50 ng/ml of doxycycline. Whole-cell lysates were resolved via SDS-PAGE. Membranes were probed using the OxPhos antibody cocktail against nuclear-encoded proteins of complexes V (ATP5A) and I (NDUFB8) and mtDNA-encoded complex IV protein MTCO2. SDHA

and SDHB were used as loading controls ($n = 1$). **C** Sucrose gradient centrifugation analysis to assess mitoribosome sedimentation patterns in GTPBP8[KO1] and GTPBP8[KO2] compared to HEK293, GTPBP8[RESCUE_wt] and GTPBP8[RESCUE_S124A]. Mitochondrial lysates were loaded onto 10-30 % isokinetic sucrose gradients and obtained fractions were analyzed via western blotting. Membranes were probed for mt-LSU MRP, mL37, and mt-SSU MRP, uS15m ($n = 3$ independent experiments). **D** In vivo translation assay of GTPBP8[KO1] and GTPBP8[KO2] cell lines compared to HEK293, GTPBP8[RESCUE_wt] and GTPBP8[RESCUE_S124A]. Cells were labeled with a [35S]-methionine and cysteine mix upon cytosolic translation inhibition. Whole cell lysates were resolved by SDS-PAGE and the signal was later detected via autoradiography. The loading was determined using Coomassie staining ($n = 1$ independent experiment). Source data are provided as a Source Data file.

initiation complex 1 (mtPIC-1) (State 3)[40,44], which includes mtIF3, and 21 % of the particles resembling the previously described mitochondrial initiation complex (IC) (State 4)[40], which includes mtIF2 and the fMet-tRNA[Met] (Fig. 6B). Of note, in the mt-SSU State 3, one unassigned extra density was found in proximity to the mtIF3 C-terminal domain, which likely corresponds to the mtIF3 N-terminal domain. In State 2, 3, and 4, adjacent to mS27, another unassigned extra density has been identified, which is linked to mS27 overexpression, as it is also present in mt-SSU particles purified from other HEK293 cell line overexpressing mS27 (data not shown) (Fig. 6B). Nevertheless, the low resolution of these regions did not allow us to build an atomic model for these densities.

Together, the absence of mature mt-LSU particles and the presence of mature mt-SSU structures in ~60% of the mt-SSU population that is retained in the GTPBP8 knock-out cells suggest that GTPBP8 may play a more critical role in the maturation and stability of the mitoribosomal large subunit.

## GTPBP8 is an RNA-binding protein that interacts specifically with the 16 S mt-rRNA

The cryo-EM structures of the mitochondrial ribosomal subunits in GTPBP8 knock-out cells led us to hypothesize that the observed mitoribosome assembly defect stems from the impairment in mt-LSU maturation. As we found no direct interaction between GTPBP8 and mt-LSU MRPs (Fig. 1 and Supplementary Fig. 2), we speculated on the possibility that GTPBP8 might be binding to RNA instead. To test this, we performed fPAR-CLIP[45] to detect RNA targets of GTPBP8::FLAG, using a Tet ON expression system in HEK293 T-REx stable cell line. After UV crosslinking GTPBP8 ribonuleoproteins (RNPs) were treated with RNase, followed by ligation of a fluorescent adapter to the RNase-protected RNA footprint bound by GTPBP8. Then, combined fluorescent imaging and western blot analysis enabled detection and isolation of GTPBP8 RNPs, migrating at ~ 45 kDa (Supplementary Fig. 9A). GTPBP8-bound RNA footprints were then further processed and sequenced.

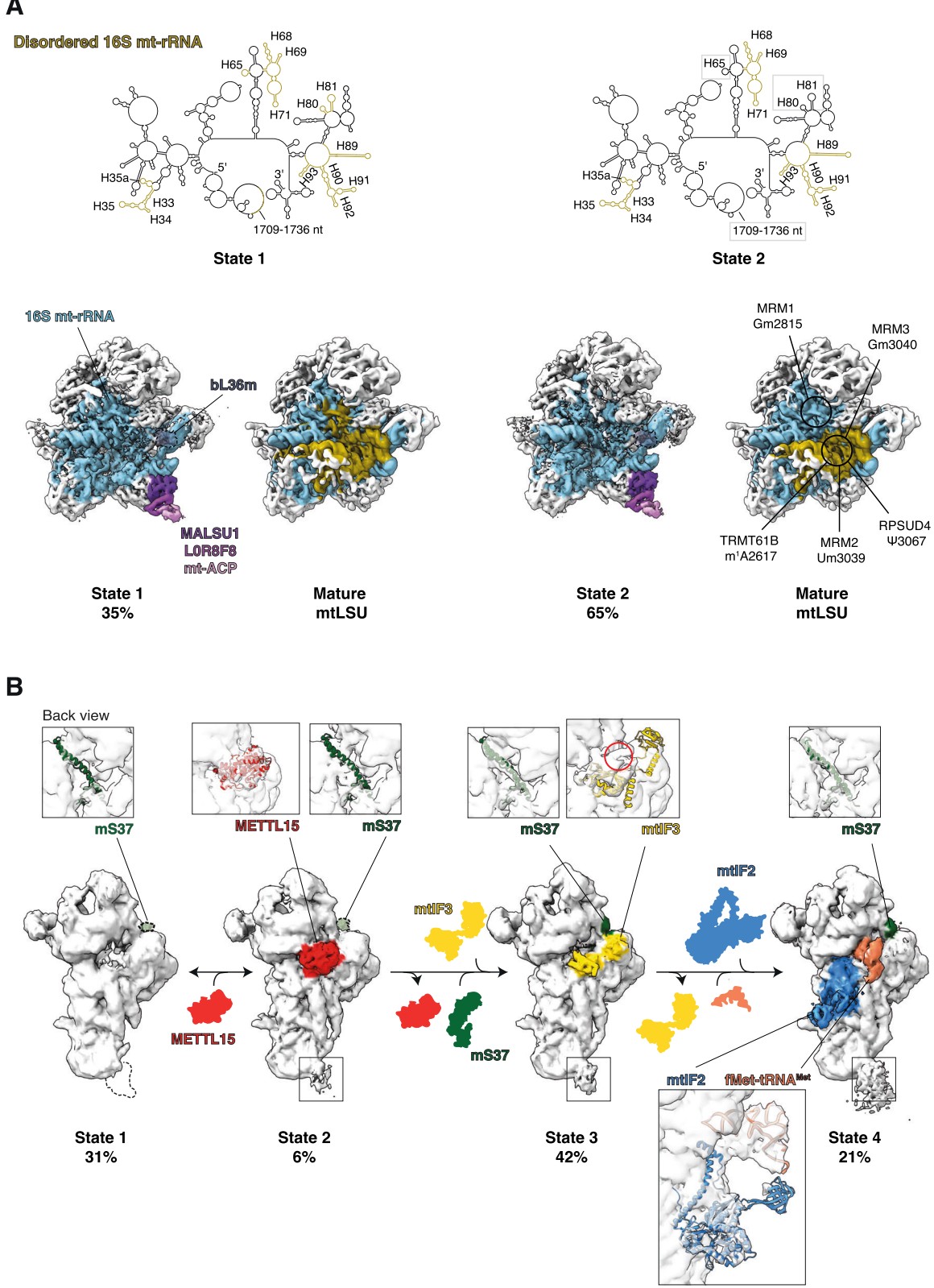

The median number of reads per GTPBP8 binding sites in mitochondrial-encoded transcripts was on average 50-fold higher than along nuclear encoding transcripts ($p = 1.74 \times 10^{-6}$), confirming GTPBP8 mitochondrial RNA binding (Supplementary Fig. 9B). While mitochondrial transcripts are known to be at high abundance in the cell, the specificity of binding to mitochondrial RNA was also confirmed by manual comparison to extremely abundant cytoplasmic mRNAs encoding for

highly expressed housekeeping proteins. RNR2 (16 S mt-rRNA) binding sites topped GTPBP8 binding at intensity ~100 fold higher relative to the mitochondrial median (Supplementary Fig. 9B, Fig. 7A). To test the specificity of GTPBP8 binding to the 16 S mt-rRNA, we used a comparative approach. First, we performed fPAR-CLIP to AUH::V5, stably expressed by an identical Tet-ON system and HEK293 cell line. AUH was shown to be a mitochondrial RNA-binding protein, interacting with AU-rich RNA

**Fig. 6 | mt-LSU and mt-SSU assembly intermediates accumulate in GTPBP8 knock-out cells. A** Upper panel. The mature secondary structure of the 16 S mt-rRNA is represented and the disordered helices of each mt-LSU sub-assembly state in GTPBP8 knock-out cells are highlighted in yellow. The gray box indicates the presence of helices H80-81, H65, and 1709-1736 nt in State 2 as compared to State 1. Lower panel. Cryo-EM densities are shown for each assembly state (State 1 and 2) and the proportion of each population is indicated as a percentage. The model-based surface of bL36m is shown in dashed lines and transparent color in both assembly states where it is lacking (PDB: 5OOL[42]). A comparison of each state with the mature mt-LSU (EMD:2762) highlights the regions of 16 S mt-rRNA which are disordered under GTPBP8 depletion (yellow). In the right panel, 16 S mt-rRNA modifications are indicated. The densities were Gaussian filtered in ChimeraX for

easier visualization. **B** Cryo-EM densities of mt-SSU populations are shown for each state and the proportion of each population is indicated as a percentage. Model-based surfaces of mS37 are shown in dashed lines and transparent color in the states where it is lacking. Density maps are highlighted in color in the states where METTL15, mS37, mtIF3, mtIF2, and fMet-tRNA[Met] are present (PDB: 6RW5[44]). For each state, zoomed-in panels of mS37, METTL15, mtIF3, mtIF2, and fMet-tRNA[Met] maps and models are represented (PDB: 6RW5[44]; 7PNX[40]). The red circle highlights the unassigned density close to the mtIF3 C-terminal domain in State 3. The black squares at the tip of the mt-SSU tail in States 2, 3, and 4 highlight the unassigned extra density in proximity of mS27::FLAG. In State 1, the absence of this extra density is shown as a dashed shape. The densities were Gaussian filtered in ChimeraX for easier visualization.

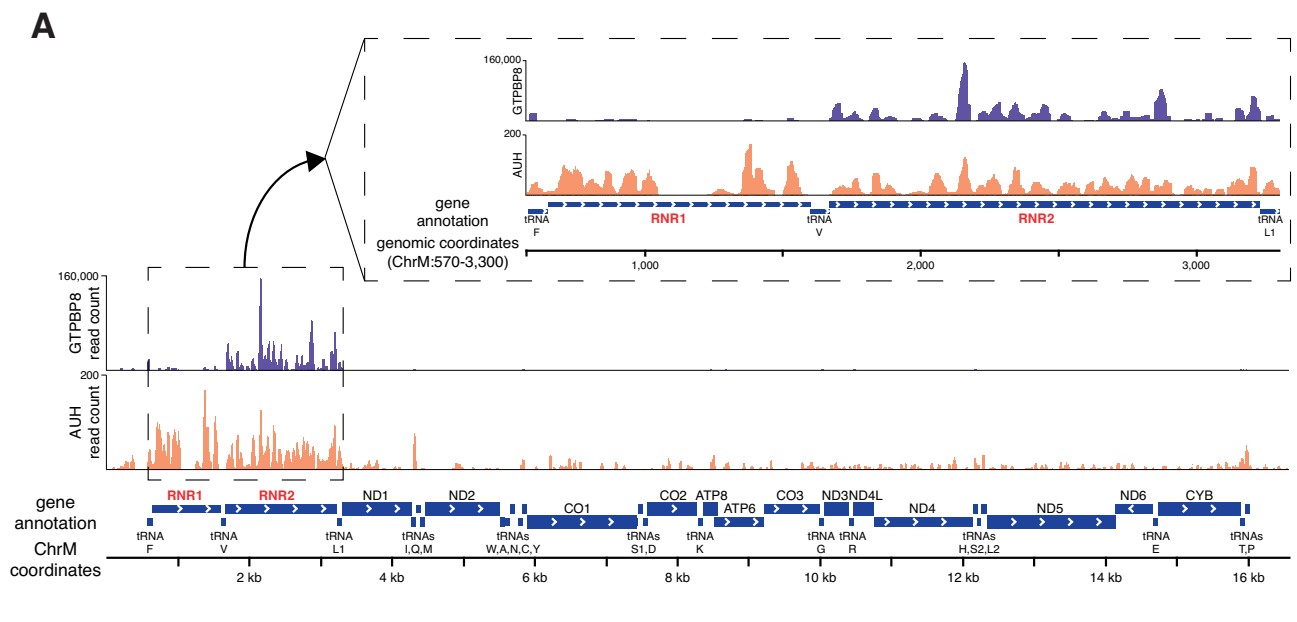

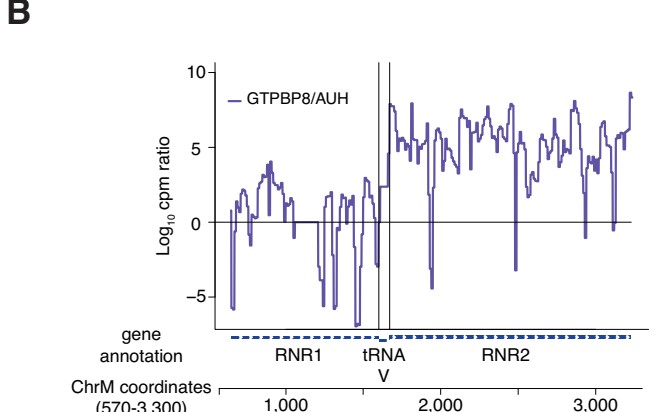

**Fig. 7 | GTPBP8 interacts with the 16 S mt-rRNA. A** Genome viewer visualization of GTPBP8 (purple) and AUH (orange) sequenced RNA footprints density along the mitochondrial chromosome (bottom) and the rRNAs (RNRs) encoding locus (top, enlarged inset). **B** The ratio between GTPBP8 and AUH sequenced RNA footprint read counts per million reads (cpm) along the 12 S (RNR1) and 16 S (RNR2) mt-rRNA loci.

elements[46]. Second, we used the 12 S mt-rRNA (RNR1) component as a reference since it is known to exist at similar levels to the 16 S mt-rRNA, and equally interact with non-specific RBPs within mitochondria (Fig. 7A, B). In line with this, GTPBP8-associated RNA almost exclusively aligned with the 16 S mt-rRNA (RNR2) region, while AUH-bound RNA aligned throughout the mtDNA, and at similar levels to RNR1 and RNR2 (Fig. 7A). To quantify the differences in AUH and GTPBP8 binding to 12 S and 16 S mt-rRNAs, we computed and plotted the ratio of binding (Fig. 7B). As shown, GTPBP8 interacts with the 16 S mt-rRNA sequence at a

$10^6$-scale fold higher intensity relative to AUH, while the ratio between the protein interactions drops to 1 immediately at the Valine tRNA bordering 16 S and 12 S mt-rRNAs, and along the 12 S mt-rRNA sequence. Given intrinsic RNase-availability patterns and other sequence and structure parameters of 16 S mt-rRNA strongly affecting its sequencability, we could not pinpoint a single GTPBP8 binding site along the 16 S mt-rRNA. Thus, this aspect requires future investigation. In summary, these data strongly indicate that GTPBP8 is an RNA-binding protein that specifically interacts with the 16 S mt-rRNA.

## Discussion

In recent years, significant progress has been made in elucidating the structural aspects of the late stages of assembly of both mt-LSU and mt-SSU. These breakthroughs have provided valuable insights into the functions of previously characterized assembly factors and identified new factors involved in this process[4]. However, despite these advancements, our knowledge still contains considerable gaps, particularly regarding the early stages of assembly. The complexity of this intricate process suggests the existence of additional assembly factors that remain unidentified.

In this study, we aimed to shed light on the role of a mitochondrially-localized, uncharacterized GTPBP known as GTPBP8. Our study demonstrated its significance in mitochondrial function and suggests its involvement in the mitochondrial large subunit assembly process.

GTPBP8 is the human homolog of bacterial EngB, which has been implicated in the maturation of the 50 S subunit in several studies[25–28]. The EngB family of GTPBPs comprises small proteins of <200 amino acids which consist of the G-domain and a short C-terminal α-helix (Supplementary Fig. 1A). GTPBP8 and *E.coli* EngB share ~35% identity, with GTPBP8 exhibiting an extra - 90 amino-acid-long N-terminal mitochondrial targeting sequence. Interestingly, in *T. brucei*, EngB cryo-EM structure in complex with the mitoribosomal large subunit has been resolved, showing that it is involved in the late steps of the assembly[32]. In particular, *T. brucei* EngB possesses a unique CTE (~400 amino acids) (Supplementary Fig. 1A) specific to this organism, which mediates its binding to the 12 S mt-rRNA and assembly factor Mtg1. As a result, it is anticipated that the bacterial and human homologs, which are devoid of this domain, may carry out their function in different ways.

To explore the potential binding of GTPBP8 to the mitoribosome in the mammalian system, we employed two distinct methods for interactome analysis: FLAG-IP and BioID. In addition, for the BioID experiment, we created cell lines overexpressing GTPBP8 with a mutation in the G-domain (S124A) to facilitate stabilization of GTP-hydrolysis-dependent interactions of GTPBP8. Surprisingly, our findings did not indicate enrichment in mt-LSU or mt-SSU MRPs. Instead, we observed an enrichment of proteins associated with RNA metabolism and mitoribosome assembly (Fig. 1 and Supplementary Fig. 2). Of note, no difference in the enrichment analysis was observed between the pull-downs performed with wild-type GTPBP8-BirA* and its mutated version (GTPBP8^{S124A}-BirA*). However, complementation experiments confirmed the functionality of the overexpressed GTPBP8 but not GTPBP8^{S124A} (Fig. 5). These results suggest that the S124A mutation interferes with GTPBP8 hydrolysis but not with GTPBP8 binding, which may occur when GTPBP8 is in the GTP-bound state. Overall, our observations still indicate a plausible role for GTPBP8 in mitoribosome biogenesis. However, it is essential to acknowledge that the absence of direct interaction with the mitoribosome might be attributed to experimental limitations.

Alongside conducting overexpression studies, we also generated GTPBP8 knock-out cells. In line with findings from other models of mitoribosome assembly deficiency, the knock-out of GTPBP8 in our study led to profound impairment in mitochondrial function. This was evident through inhibited translation (Fig. 3A–B) and a significant OxPhos defect (Fig. 2C–E) without underlying general transcriptional defect (Fig. 3C). These findings underscore the critical role of GTPBP8 in maintaining proper mitochondrial function and highlight its involvement in mitoribosome-related processes. However, unlike the knock-out or downregulation of other mitoribosome assembly factors, which typically have negative effects on a specific subunit[7,17,34,38], GTPBP8 ablation has a remarkably strong detrimental effect on both the mt-LSU and mt-SSU. This is evident from the noticeable downregulation of steady-state levels of MRPs from both subunits, as well as reduced levels of 16 S and 12 S mt-rRNAs (Fig. 4A–C and Supplementary Fig. 3D).

Among all the MRPs, mS37 has been proposed to be important for the coordination between the mt-LSU and mt-SSU assembly, as its steady-state levels are consistently reduced in models of mt-LSU deficiency[40,41]. Indeed, in both biological replicates of GTPBP8 knockouts, mS37 is the most downregulated MRP of the mt-SSU, as assessed both via western blotting and mass spectrometry (Fig. 4A–C). Aligning with this finding, our cryo-EM data revealed the presence of the stable mt-SSU assembly intermediates lacking mS37, while the rest of the resolved mt-SSU complexes (~60 %) were fully matured and associated with translation initiation factors (Fig. 6B). In contrast, GTPBP8 knock-out cells exhibited only mt-LSU assembly intermediates, with no fully matured mt-LSU subunits detected (Fig. 6A). Notably, the mt-LSU intermediates accumulated in GTPBP8 knock-out cells are among the earliest mt-LSU intermediates identified thus far, as they display partly disordered 16 S mt-rRNA domain I and II (H33-35a) and completely unstructured domain IV and V (H67-71, H89-93, H80-81) (Fig. 6A). An early defect in mt-LSU assembly in GTPBP8 knock-out cells may thus be the explanation for the manifestation of more severe effects on the mt-SSU.

The specific participation of GTPBP8 in mt-LSU biogenesis was further corroborated by our fPAR-CLIP analysis which showed that GTPBP8 interacts exclusively with 16 S mt-rRNA (Fig. 7). However, we were not able to determine the exact binding site on the rRNA. Future studies are needed to determine whether GTPBP8 plays a role in folding of specific domains of the 16 S mt-rRNA or if it assists in the insertions of rRNA posttranscriptional modifications. In support of our findings, another study on GTPBP8 has recently been published, suggesting its role in mitoribosome assembly, however, the mechanistic details of this function remain to be determined[47].

Collectively, our data suggest a direct involvement of GTPBP8 in mt-LSU maturation, as GTPBP8 binds to the 16 S mt-rRNA and the mt-LSU fails to reach full maturity under GTPBP8 depletion. On the other hand, the impact on mt-SSU maturation seems to be indirect as, despite a substantial reduction in mt-SSU MRPs, a portion of the remaining mt-SSU population is mature and capable of engaging in translation initiation. This phenomenon likely serves as a compensatory mechanism to counteract the severe OxPhos defect resulting from GTPBP8 depletion.

To fully understand GTPBP8's function, further investigations are necessary to unravel its exact role in 16 S mt-rRNA maturation. Moreover, the observed detrimental effects of the absence of GTPBP8 on the mt-SSU raise intriguing questions about the underlying mechanisms. Additional research focused on the early stages of mitoribosome assembly will be essential to shed light on the origin of these effects and to clarify the precise role of GTPBP8 in coordinating the formation and maturation of both subunits. By gaining a deeper understanding of GTPBP8's interactions and functions, we can advance our knowledge of mitoribosome biogenesis and its implications for mitochondrial translation and overall mitochondrial function.

## Methods

### Flp-In T-Rex stable mammalian cell lines generation and maintenance

Stable mammalian cell lines that enable doxycycline-inducible expression of C-terminally FLAG-tagged GTPBP8 (GTPBP8::FLAG) were generated using the Flp-In T-Rex human embryonic kidney 293 (HEK293) cell line (Invitrogen). The internal ID 29083 of the hORFeome Database's GTPBP8 cDNA was used to clone it into pcDNA5/FRT/TO. Supplementary Table 1 contains a list of the primers that were utilized to clone GTPBP8, GTPBP8 mutant, and mS27 protein. GTPBP8-BirA*-HA and GTPBP8^{S124A}-BirA*-HA (mitochondrial targeting sequence from human COX8a) were also cloned into pcDNA5/FRT/TO. HEK293 cells were cultured in DMEM (Dulbecco's modified eagle medium) supplemented with 10% (v/v) tetracycline-free fetal bovine serum (FBS), 2 mM Glutamax (Gibco), 1 x Penicillin/Streptomycin (Gibco), 50 μg/ml

uridine, 10 μg/ml Zeocin (Invitrogen) and 100 μg/ml blasticidin (Gibco) at 37 °C under 5% $CO_2$ atmosphere. Before transfection, cells were seeded in a 6-well plate and cultured in culture media devoid of selective antibiotics. Cells were transfected with the pcDNA5/FRT/TO-GTPBP5::FLAG and pOG44 constructs using Lipofectamine 3000 following the manufacturer's recommendations. Forty-eight hours after transfection, hygromycin (100 μg/ml, Invitrogen) and blasticidin (100 μg/ml) were added to the culture medium for the selection of positive clones. After two to three weeks, selected colonies were picked and cultured as single clonal populations. Cells were treated with 50 ng/ml doxycycline to confirm the inducible expression of GTPBP8, and 24 or 48 h later, cell pellets were collected for Western Blot examination.

For cell growth measurements, cells were plated in 6-well plates at a density of $2 \times 10^4$ and cultured in glucose-free DMEM supplemented with 0.9 g/L galactose, 10% (v/v) FBS, 1 mM sodium pyruvate, 2 mM Glutamax, 1 x Penicillin/Streptomycin and uridine. EVE™ Automated Cell Counter (NanoEnTek) was used to measure cell count at 24-h intervals.

## GTPBP8 CRISPR/Cas9 knock-out generation

To generate GTPBP8 knock-out clones, the CRISPR/Cas9 system was employed as described[48]. In order to induce out-of-frame deletions, exon 1 of the *GTPBP8* gene was targeted using two pairs of gRNAs which were cloned into the pSpCas9(BB)−2A-Puro (pX459) V2.0 vector. Afterward, HEK293 cells were transfected with the pX459 vectors using Lipofectamine 3000 following the manufacturer's recommendations. Puromycin at a final concentration of 1.5 μg/ml was added to the culture medium for 48 h to select the transfected cells. Cells were then subjected to single-cell dilution on 96-well plates. The deletion of GTPBP8 in selected clones was verified by Sanger sequencing and by Western blotting.

## Immunocytochemistry analysis

An immunocytochemistry (ICC) study was performed using the human 143B osteosarcoma (HOS) cell line. DMEM supplemented with 10% (v/v) FBS, 2 mM Glutamax, and 1 x Penicillin/Streptomycin was used to culture the cells prior to the experiment. 143B cells were transiently transfected with pcDNA5/FRT/TO-GTPBP8::FLAG with Lipofectamine 3000, according to the manufacturer's instructions. Fixed cells were utilized for the ICC as described in previous work[49]. TOM20 was used as a marker of the mitochondrial network. Images were acquired using a Zeiss LSM800 confocal microscope.

## Cell fractionation

Cell fractionation was carried out as depicted in ref. 50 devoid of the sucrose gradient step.

## Western blotting

GTPBP8::FLAG, GTPBP8^RESCUE_wt and GTPBP8^RESCUE_S124A cell lines were induced with either 1 ng/ml or 50 ng/ml of doxycycline 48 h prior to cell pelleting. Cell lysates were used for the study of protein steady-state levels. Samples were resuspended in 1× NuPAGE LDS sample buffer (Thermo Fisher) and 100 mM dithiothreitol (DTT), heated at 95 °C and subjected to SDS-PAGE using NuPAGE 1× MES (Thermo Fisher) running buffer. Proteins were later blotted onto PVDF membranes (Millipore) in transfer buffer (40 mM glycine, 50 mM Tris, and 20% methanol) at 4 °C for 1 h and 30 min at 300 mA. Blocking solution of 5% (w/v) milk in 1x phosphate buffered saline containing 0.1% Tween 20 (PBS-T) was applied to the membranes for 1 h at room temperature (RT). Immunodetection was performed using specific primary antibodies (Supplementary Table 2) diluted in 5% milk PBS-T incubating overnight at 4 °C. Following three washes with PBS-T, membranes were incubated with HRP-conjugated secondary antibodies diluted in 5% milk PBS-T for 1 h at RT. After three more washes in PBS-T, the signal

was visualized using ECL (Bio-Rad). Detected bands were quantified using Image Lab software (Bio-Rad).

## Mitochondrial isolation

Isolation of mitochondria was performed as previously described in Rorbach et al.[51]. In brief, collected cells were resuspended in MSE buffer (150 mM D-mannitol, 100 mM Tris pH 7.4, 1 mM EDTA) supplemented with 0.1 % (w/v) bovine serum albumin (BSA). Several homogenization steps (Homgenplus Homogenizer, Schuett-biotec) were employed to disrupt the cells. Mitochondria were first separated after different rounds of differential centrifugation at 400 × g for 10 min at 4 °C and then harvested at 11,000 × g for 10 min at 4 °C. The mitochondrial pellet was later resuspended in MSE buffer and loaded on a step gradient (1–1.5 M sucrose, 20 mM Tris pH 7.4, 1 mM EDTA) and centrifuged at 106,880 × g for 1 h at 4 °C (SW41-Ti rotor, Beckman Coulter). Mitochondria layered at the interface of the two solutions were collected and 10 mM Tris HCl pH 7.4 in 1:1 ratio was added. The final mitochondrial pellet was obtained after subsequent centrifugation at 11,000 × g for 10 min at 4 °C, then it was resuspended in freezing buffer (300 mM trehalose, 10 mM Tris pH 7.4, 10 mM KCl, 1 mM EDTA, 0.1% BSA), snap-frozen in liquid nitrogen and stored at −80 °C.

## Sucrose density gradient ultracentrifugation

GTPBP8::FLAG, GTPBP8^RESCUE_wt, and GTPBP8^RESCUE_S124A cell lines were induced with 50 ng/ml of doxycycline for 48 h and collected at a confluency of 90% prior to mitochondrial isolation. Mitochondrial pellets were lysed in lysis buffer (10 mM Tris-HCl pH 7.5, 50 mM KCl, 20 mM MgCl2, 1x PIC, 260 mM sucrose, 1% Triton X-100) containing freshly added RNase Block and 1 mg of the protein content was loaded onto a continuous sucrose gradient (10–30% (w/v), 20 mM Tris-HCl pH 7.5, 50 mM KCl, 20 mM MgCl2, 1x PIC) and centrifuged for 15 h at 79,000 × g at 4 °C (Beckman Coulter SW41-Ti rotor). A total of 25 fractions was collected, each with a volume of 450 μl, and 15 μl of each fraction was used for the Western Blot analysis. Fractions 1 and 2, and fractions 18 and 19, were mixed and resolved simultaneously.

Purification of the mt-LSU sub-assembly intermediates from GTPBP8^KO1 for the cryo-EM experiment was performed as described in ref. 13 with some modifications. After mitochondrial isolation, crude mitochondria were further purified via differential centrifugation onto a sucrose step gradient (1-1.5 M sucrose, 20 mM Tris-HCl pH 7.5, 1 mM EDTA) at 107,170 × g for 1 h at 4 °C (Beckman Coulter SW41-Ti rotor). Mitochondria were collected from the interphase between the 1 M and the 1.5 M solution, resuspended in 10 mM Tris-HCl pH = 7.5 at a ratio of 1:1, and pelleted via centrifugation. The mitochondrial pellet was later resuspended in mitochondrial freezing buffer (200 mM trehalose, 10 mM Tris-HCl pH 7.5, 10 mM KCl, 0.1% bovine serum albumin, 1 mM EDTA), snap frozen in liquid nitrogen, and stored at −80 °C. Prior to the experiment, mitochondria were lysed at 4 °C for 20 min (25 mM HEPES-KOH pH = 7.5, 20 mM Mg(OAc)2, 100 mM KCl, 2% (v/v) Triton X-100, 2 mM dithiothreitol (DTT), 1×cOmplete EDTA-free protease inhibitor cocktail (Roche), 40 U/μl RNase inhibitor (Invitrogen)) and then subjected to sucrose cushion ultracentrifugation method (0.6 M sucrose, 25 mM HEPES-KOH pH = 7.5, 10 mM Mg(OAc)2, 100 mM KCl, 0.5% (v/v) Triton X-100, 2 mM DTT) at 259.000 × g for 2 h and 30 min at 4 °C (Beckman Coulter TLS55 rotor). The obtained pellet was resuspended in ribosome resuspension buffer (25 mM HEPES-KOH pH = 7.5, 10 mM Mg(OAc)2, 100 mM KCl, 0.05% DDM, 2 mM DTT) and centrifuged at 15,871 × g for 10 min at 4 °C. The supernatant was subsequently loaded onto a linear 10-30 % (w/v) sucrose gradient (25 mM HEPES-KOH pH = 7.5, 10 mM Mg(OAc)2, 100 mM KCl, 0.05% DDM, 2 mM DTT) and centrifuged at 93,000 × g at 4 °C for 2 h and 15 min (Beckman Coulter TLS55 rotor). The mt-LSU fractions were collected and concentrated via buffer exchange method (25 mM HEPES-KOH pH = 7.5, 10 mM Mg(OAc)2, 100 mM KCl) using Vivaspin 500 centrifugal concentrators.

## Flag-immunoprecipitation

Prior to mitochondrial isolation, 70 % confluent cells were induced with 50 ng/ml of doxycycline for 48 h. Pelleted mitochondria were resuspended in lysis buffer (Sigma-Aldrich, St. Louis, MO) containing RNase Block Ribonuclease Inhibitor (Agilent), 1x PIC (cOmplete, Mini, EDTA-Free, Protease Inhibitor Cocktail, Roche) and 5 mM $MgCl_2$. Protein quantification was performed using a Qubit protein quantification assay (Thermo Fisher). Immunoprecipitation of GTPBP8::FLAG was performed following the manufacturer's instructions of the FLAG-immunoprecipitation (IP) kit (Sigma-Aldrich). Wash and elution buffers were supplemented with 5 mM $MgCl_2$. For the elution step, 3X FLAG peptides were used and samples were shaken for 30 min at 4 °C.

Purification of the mt-SSU sub-assembly intermediates via FLAG-IP from GTPBP8[KO1] cells transfected with mS27::FLAG protein for the cryo-EM experiment was performed as described in ref. 13. Mitochondria were purified as described in the "Sucrose density gradient ultracentrifugation" method paragraph. In brief, the mitochondrial pellet was lysed in cold for 20 min (25 mM HEPES-KOH pH = 7.5, 20 mM Mg(OAc)$_2$, 100 mM KCl, 2% (v/v) Triton X-100, 0.2 mM DTT, 1 × cOmplete EDTA-free protease inhibitor cocktail (Roche), 40 U/µl RNase inhibitor (Invitrogen)) and subsequently centrifuged at 5,000 × g for 5 min at 4 °C. The mitolysate was then incubated with ANTI-FLAG M2-Agarose Affinity Gel (Sigma-Aldrich), previously equilibrated (25 mM HEPES-KOH pH = 7.5, 5 mM Mg(OAc)$_2$, 100 mM KCl, 0.05% DDM), at 4 °C for 3 h. The sample was later centrifuged at 5000 × g for 1 min at 4 °C and the beads were washed three times prior to elution (25 mM HEPES-KOH pH = 7.5, 5 mM Mg(OAc)$_2$, 100 mM KCl, 0.05% DDM). The elution step was performed using the elution buffer (25 mM HEPES-KOH pH = 7.5, 5 mM Mg(OAc)$_2$, 100 mM KCl, 0.05% DDM, 2 mM DTT) supplemented with 3 × FLAG Peptide (Sigma-Aldrich) provided by the kit for about 40 min at 4 °C shaking.

## BioID

The BioID experiment was carried out following the published protocol[37] with some modifications. Prior to the experiment, 70% confluent cells were induced with 1 ng/ml of doxycycline for 9 h, 6 h of which they were treated with 50 µM of biotin. Afterward, mitochondria were isolated and lysed in 1 ml lysis buffer (50 mM Tris pH 7.4, 500 mM NaCl, 0.2 % SDS, 1x PIC, 1 mM DTT, 0.4 % Triton X-100) at 4 °C for 30 min. The mitolysates were collected after centrifugation at 16,500 × g for 10 min at 4 °C and incubated with Dynabeads MyOne Streptavidin C1 (Thermo Fisher) at 4 °C rotating overnight. Beads were later washed two times with wash buffer 1 (2% SDS), two times with wash buffer 2 (0.1 % sodium deoxycholate, 1 mM EDTA pH 8, 1 % Triton X-100, 50 mM HEPES pH 7.5), and 5 times with 50 mM Tris pH 7.4. Samples were then eluted (2 M Urea, 5 ng/µl trypsin gold (Promega, cat. no. V5280, resuspended in 50 mM acetic acid), 50 mM Tris pH 7.4, 1 mM TCEP (Thermo Fisher)) shaking at RT for 30 min.

## Preparation of peptides from FLAG-IP for mass spectrometry

Eluates obtained from GTPBP8::FLAG and mtLuc::FLAG immunoprecipitation ($n = 4$) were resuspended with 300 µl of 0.6 M Guanidium chloride buffer (5 mM Tris-HCl pH 7.5, 0.1 mM Tris(2-carboxyethyl) phosphine (TCEP; ThermoFisher), 0.5 mM 2-chloroacetamide (CAA) (Merck, cat. no. 8024120100)). Overnight digestion was performed using 300 ng of trypsin gold followed by the peptide desalting step using home-made StageTips (Empore Octadecyl C18, 3 M[52];). Peptides were eluted with 100 µl of 60% acetonitrile / 0.1% formic acid buffer and subsequently dried using a SpeedVac Vacuum Concentrator. Prior to mass spectrometry analysis, peptides were resuspended in 0.5% formic acid.

## Preparation of peptides from BioID for mass spectrometry

After the elution step, 5 mM CAA was added and samples were incubated overnight at 37 °C to ensure complete tryptic digestion (1:50 w/w). The next steps were carried out as described for the FLAG-IP samples.

## Preparation of peptides from mitolysates for mass spectrometry

Mitochondrial pellets (50 µg) were resuspended in 100 µl of GuHCl/Tris pH 8.0 solution (6 M GuHCl solution in 100 mM Tris-HCl pH 8.0) and sonicated twice for 5 min (10 s ON/OFF cycles). Then, they were centrifuged at maximum speed for 10 min at RT. At this stage, proteins were reduced using 5 mM DTT for 30 min at 55 °C, cooled down briefly, and alkylated with 300 mM CAA for 15 min at RT in the dark. During the alkylation step, the samples' protein content was quantified by BCA assay. Samples were later diluted 10 times using 50 mM Tris pH 8 to stop the alkylation reaction. Protein digestion was performed overnight at 37 °C using trypsin (1:50 w/w) (Pierce, trypsin protease MS-grade, Thermo Fisher) according to protein quantification measurements. Afterward, trypsin was inactivated with 1.2% formic acid and samples were spun down at 3000 × g for 10 min at RT. Peptides were then purified using Pierce Peptide Desalting Spin columns (Thermo Fisher) equilibrated and washed according to the manufacturer's recommendations. Elution was performed using 50% acetonitrile / 0.5% formic acid buffer and peptides were dried using a SpeedVac Vacuum Concentrator. Prior to mass spectrometry analysis, peptides were resuspended in 0.5% formic acid.

## LC-MS/MS analysis

IP samples were analyzed using data dependent acquisition (DDA). Peptides were separated on a 25 cm, 75 µm internal diameter PicoFrit analytical column (New Objective) packed with 1.9 µm ReproSil-Pur 120 C18-AQ media (Dr. Maisch,) using an EASY-nLC 1200 (Thermo Fisher Scientific). The column was maintained at 50 °C. Buffer A and B were 0.1% formic acid in water and 0.1% formic acid in 80% acetonitrile. Peptides were separated on a segmented gradient from 6% to 31% buffer B for 45 min and from 31% to 50% buffer B for 5 min at 200 nl / min. Eluting peptides were analyzed on a QExactive HF mass spectrometer (Thermo Fisher Scientific). Peptide precursor m/z measurements were carried out at 60000 resolution in the 300 to 1800 m/z range. The ten most intense precursors with charge states from 2–7 only were selected for HCD fragmentation using 25% normalized collision energy. The m/z values of the peptide fragments were measured at a resolution of 30000 using a minimum AGC target of 2e5 and 80 ms maximum injection time. Upon fragmentation, precursors were put on a dynamic exclusion list for 45 s.

BioID and mitoproteome samples were analyzed using data-independent acquisition (DIA). Peptides were separated on a 15 cm, 75 µm internal diameter packed emitter column (Coann emitter from MS Wil, Poroshell EC C18 2.7 micron medium from Agilent) using an EASY-nLC 1200 (Thermo Fisher Scientific). The column was maintained at 50 °C. Buffers A and B were 0.1% formic acid in water and 0.1% formic acid in 80% acetonitrile, respectively. Peptides were separated on a gradient from 4% to 30% buffer B for 19 min at 400 nl / min, followed by a higher organic wash. Eluting peptides were analyzed on a QExactive HF mass spectrometer (Thermo Fisher Scientific) in DIA mode. Peptide precursor m/z measurements were carried out at 120000 resolution in the 400–800 m/z range followed by 29 DIA scans with an isolation width of 14 Th and a resolution of 15000. MS1 and DIA MS2 scans were recorded in centroid mode.

## Mass spectrometry analysis: protein identification and quantification

GTPBP8 IP raw data were analyzed with MaxQuant version 1.6.1.0[53] using the integrated Andromeda search engine[54]. Peptide fragmentation spectra were searched against the canonical sequences of the human reference proteome (proteome ID UP000005640, downloaded September 2018 from UniProt). Methionine oxidation and

protein N-terminal acetylation were set as variable modifications; cysteine carbamidomethylation was set as a fixed modification. The digestion parameters were set to "specific" and "Trypsin/P," The minimum number of peptides and razor peptides for protein identification was 1; the minimum number of unique peptides was 0. Protein identification was performed at a peptide spectrum match and protein false discovery rate of 0.01. The "second peptide" option was on. Successful identifications were transferred between the different raw files using the "Match between runs" option. Label-free quantification (LFQ)[55] was performed using an LFQ minimum ratio count of two.

BioID and mitoproteome data were analyzed with Spectronaut 16.2 (Biognosys) using default parameters against the reference proteome for humans, UP000005640, downloaded in September 2018. Methionine oxidation and protein N-terminal acetylation were set as variable modifications; cysteine carbamidomethylation was set as a fixed modification. The digestion parameters were set to "specific" and "Trypsin/P," with two missed cleavages permitted. Protein groups were filtered for at least two valid values in at least one comparison group and missing values were imputed from a normal distribution with a down-shift of 1.8 and standard deviation of 0.3. Differential expression analysis was performed using limma, version 3.34.9[56], in R, version 3.4.3[57]. Mitocarta2[58] and Mitocarta3[59] annotations were added using the primary gene name and the first of the gene name synonyms of the oldest Uniprot ID with the highest number of peptides. MitoCarta3 pathway enrichment analysis was performed using WebGestalt[60].

### [35S]-labeling of de novo mitochondrial translation

Cells were seeded in a 6-well plate to reach 80–90% confluency on the day of the experiment. GTPBP8::FLAG, GTPBP8$^{RESCUE\_wt}$ and GTPBP8$^{RESCUE\_S124A}$ were induced with 50 ng/ml of doxycycline 48 h prior to the experiment. First, two 5 min-long washings in 2 ml of DMEM devoid of methionine and cysteine were carried out. After that, cells were incubated for 20 min at 37 °C in 1 ml of fresh Cys-/Met-free DMEM, supplemented with Glutamax 100X (Gibco), Sodium Pyruvate 100X (Gibco), 10 % dialyzed FBS, and 100 μg/ml emetine (Sigma-Aldrich), the latter used to inhibit cytosolic translation. Then, 200 mCi/ml of EasyTag EXPRESS [35S] protein labeling mix (methionine and cysteine) (Perkin Elmer) was added for 30 min at 37 °C. Later, cells were washed three times with PBS at 4,000 × g for 5 min at 4 °C. Cell lysis was performed using 1x PBS-PIC with the addition of 50 units of benzonase (Life Technologies) and applying one freeze-thaw cycle overnight. Protein content was determined using BCA assay and 30 μg of cell lysates were run on Bolt 12% Bis-Tris Plus (Invitrogen) SDS-PAGE gels. Imperial Protein Stain (Thermo Fisher) for Coomassie staining first, and a fixing solution (20% methanol, 7% acetic acid, and 3% glycerol) afterward, were applied to the gels for 1 h each at RT. Eventually, gels were vacuum-dried for 2 h at 65 °C and later exposed to storage Phosphor screens and visualized with Typhoon FLA 7000 Phosphorimager.

### RNA isolation and northern blotting

Total RNA was isolated from HEK293, GTPBP8$^{KO1}$, and GTPBP8$^{KO2}$ using TRIzol (Invitrogen) according to the manufacturer's instructions. RNA was solubilised in water at 4 °C and 40 units of RNase block were added. RNA concentrations were measured using NanoDrop ND-1000 UV-Vis Spectrophotometer (Thermo Scientific).

For the study of the steady-state levels of mt-mRNAs and mt-rRNAs, 4 μg of RNA samples were resuspended in NorthernMaxTM-Gly sample loading dye (Thermo Fisher), heated for 10 min at 55 °C and cooled down on ice prior to separation on 1.2% (w/v) agarose gels with 0.7% formaldehyde and 1x NorthernMax Running Buffer (Thermo Fisher). The RNA was then transferred overnight onto Hybond-N+ membranes (Cytiva) and then UV-crosslinked (UVC 500 Crosslinker, Amersham Biosciences, 254 nm, 120 mJ/cm²) prior to activation of the membrane in hybridization buffer (Sigma) for 10 min at 65 °C.

Membranes were subsequently incubated with [32P]-radiolabelled DNA probes obtained using the Prime-It II Random Primer Labeling Kit (Agilent Technologies) according to the manufacturer's instructions. Firstly, 20 ng of DNA was mixed with 10 ul of random oligonucleotide primers and heated at 95 °C for 5 min. Following a short centrifugation, samples were incubated at RT for 5 min. The reaction was completed with the addition of 10 μl of 5× dCTP buffer, 1 μl of 5 U/μl Exo(-) Klenow enzyme, and 20–30 μCi of [α-32P]-dCTP (PerkinElmer) and incubation at 37 °C for 30 min, followed by another incubation at 95 °C for 5 min and on ice for 2 min. The probes were then purified with Amersham MicroSpin G-50 columns (Cytiva) and then added to the hybridization buffer and left incubating with the membranes at 65 °C for variable times (2 h up to overnight). Membranes were then washed three times with 1× SSC buffer (150 mM NaCl, 15 mM tri-sodium citrate, pH 7.0) containing 0.1% (w/v) SDS at 65 °C for 10 min. Finally, membranes were exposed to storage Phosphor screens and the signal was visualized using Typhoon FLA 7000 Phosphorimager.

Regarding tRNA steady-state levels analysis, 4 μg of RNA samples were resuspended in Gel loading buffer II (Invitrogen), heated at 80 °C for 3 min, and cooled down on ice prior to separation on 15% TBE-Urea Gel (Invitrogen) in 1x TBE Running Buffer (novex). The RNA was later transferred and UV-crosslinked (UVC 500 Crosslinker, Amersham Biosciences, 254 nm, 120 mJ/cm²) onto Hybond-N+ membranes (Cytiva). Membranes were then activated in hybridization buffer (Sigma) for 10 min at 65 °C and then incubated with [32P]-radiolabelled DNA probes. Probes were generated using the Riboprobe Combination System – SP6/T7 kit (Promega) according to the manufacturer's instructions and [α-32P]-rUTP (PerkinElmer) was used. After purification of the probe, the same steps as for the rRNA/mRNAs northern blotting were performed.

The list of primers designed to generate DNA templates for the probes can be found in Supplementary Table 3.

### Mitoribosome profiling

Sample preparation was performed as described in Krüger et al.[61]. The pooled sequencing data were split by the index barcodes used for each sample during the pooling PCR and converted to the FASTAQ format by bcl2fastq (Illumina). First, the common 3′ adapter sequence (TGGAATTCTCGGGTGCCAAGG) was trimmed off from the adapter reads by using Cutadapt (filter settings: max-n 0.2 -m 32 -M 40). Then, the trimmed reads were aligned to the mitochondrial genome by using the Bowtie 2 local alignment[62]. The P-site of the mapped reads was identified by Plastid (Dunn and Weissman, 2016). The offset of the P-site from the 5′ ends of reads was calibrated using the mtCOX1 stop codon peak. Periodicity was checked across all ORFs and exact P-site locations were adjusted (32: 15; 33: 15; 34:16; 35:16; 36:16; 37: 16; 38:16; 39: 18; 40:16; 41:16; 42: 16). Finally, codon count vectors for all transcripts were generated. Ratios of ribosome counts of GTPBP8$^{KO1}$ vs. control cells were calculated for each transcript. Ribosome counts on individual codons were plotted for ATP6 and ND4 transcripts.

### OxPhos complexes activity assay using BN-PAGE

After mitochondrial isolation, mitochondrial pellets were resuspended in ACNA buffer (1.5 M aminocaproic acid, 50 mM Bis-Tris pH 7) and then quantified using a Qubit protein quantification assay. 50 μg of mitochondria were lysed using 4% (w/v) digitonin on ice for 20 min and then centrifuged at maximum speed for 30 min at 4 °C. After collecting the supernatants, they were resuspended in blue native loading dye (5% Coomassie blue G, 500 mM e-amino n-caproic acid in 100 mM Bis-Tris pH 7.0) and then separated by 3–12% Blue-Native Polyacrylamide Gel Electrophoresis (BN-PAGE) (Thermo Fisher) for about 5 h using NativePAGE Running buffer kit (Thermo Fisher). In-gel activity measurements were performed for Complex I and Complex IV using buffers with their specific substrates (Complex I: 2 mM Tris-HCl, pH 7.4, 0.1 mg/mL NADH, 2.5 mg/mL iodonitrotetrazolium chloride; Complex

IV: 0.5 mg/mL DAB, 50 mM phosphate buffer pH 7.4, 1 mg/mL cytochrome c, 0.2 M sucrose, 20 μg/mL (1 nM) catalase).

## RT-qPCR

RNA extraction was performed using the RNeasy Mini kit (Qiagen) according to the manufacturer's protocol. Residual DNA contamination was removed by on-column DNase digestion (RNase-Free DNase Set, Qiagen). After RNA extraction and quantification, 1 mg of total RNA was retrotranscribed into cDNA using the High-Capacity cDNA Reverse Transcription kit (Thermo Fisher). In order to determine the steady-state levels of mt-mRNAs and mt-rRNAs, the generated cDNA was diluted 1:20 with nuclease-free water and 1% of the total was used in each 10 ml reaction. The TaqMan Universal PCR Master Mix (Applied Biosystems) and TaqMan probes labeled with FAM (6-carboxyfluorescein) reporter (Applied Biosystems) were combined and used to perform quantitative PCR using the QuantStudio 6 Flex Real-Time PCR System (Applied Biosystems). Supplementary Table 4 provides the list of all the probes used in this study. To calculate the fold change of mt-RNAs in comparison to the endogenous housekeeping control ACTB, the $2^{-DDCt}$ method was used.

## Cryo-EM data acquisition and image processing

Quantifoil Cu 300 mesh (R 2/2 geometry) grids covered with a thin carbon layer of 2 nm were used for the preparation of the cryo-EM samples (Agar Scientific). Grids were glow discharged at 20 mA for 30 s using an EMS100X easiGlow glow-discharge unit prior to loading of the sample. Three μl of the sample were later applied to the grids and vitrified at 4 °C and 100 % humidity using a Vitrobot Mk IV (Thermo Fisher Scientific) (blot 3 s, blot force 1, waiting time 30 s, 595 filter paper (Ted Pella Inc.)).

All cryo-EM data collection (Supplementary Table 6) was performed with EPU (Thermo Fisher Scientific) using a Krios G3i transmission-electron microscope (Thermo Fisher Scientific) operated at 300 kV in the Karolinska Institutet's 3D-EM facility. Images were acquired in nanoprobe 165kX EF-TEM SA mode (0.505 Å/px) with a slit width of 10 eV using a K3 Bioquantum, for 1.5 s with 60 fractions and a total fluency of 48 e − /Å². For the mt-SSU dataset, the stage tilt was set to 0°, −20° or −30°. Motion correction, dose-weighting, CTF-estimation, Fourier cropping (to 1.01 Å/px), particle picking (size threshold 300 Å, distance threshold 20 Å, using the pre-trained BoxNet2Mask_20180918 model), and extraction in 512-pixel boxes were performed on the fly using Warp[63]. Only particles from micrographs with an estimated resolution of 4 Å or better and under-focus between 0.2 and 3 μm were retained for further processing.

For the mt-LSU sample, Warp picked 404,181 particles from 10,162 micrographs. The particles were imported into CryoSPARC v4[64] for further processing. First, 2D classification was performed and the good 2D classes were selected to generate an Ab-initio volume of the LSU. A second Ab-initio volume using the particles from bad quality 2D classes was also generated. These two volumes were used for heterogeneous refinement of the entire dataset, downsampled to 3.03 Å/px. We found that this processing strategy is very efficient in filtering out the bad particles. The remaining particles (195,744) were subjected to homogeneous refinement. Finally, we performed 3D classification, with a solvent mask (but not a focus mask). This strategy revealed 6 classes as shown in Supplementary Fig. 5A and Supplementary Fig. 6. Of these 6 classes, 3 were discarded as they did not refine to a resolution beyond 6 Å. The remaining 3 were further 3D classified, using an interface focus mask, and pooled into two classes with different RNA folding (Supplementary Fig. 5A and Supplementary Fig. 6).

The LSU states 1-2 atomic models were built using the PDB 5OOM[42] as a starting model. The models were docked in the maps using UCSF ChimeraX, model-map discrepancies were re-built in Coot and finally refined using Phenix Real Space Refine.

For the mt-SSU sample, Warp-picked 437,172 particles were imported into CryoSPARC v4 for further processing, with a similar strategy to mt-LSU. First, 2D classification was performed and good 2D classes were used to generate an Ab-initio volume of the mt-SSU. Two additional volumes were generated from the bad 2D classes. The three volumes were used in the heterogeneous refinement of the entire WARP-picked particles to filter out the bad particles. The same was done for two datasets collected with −20° and −30° stage tilt. The particles were then combined and subjected to homogeneous refinement (195,904). Finally, we performed 3D classification with a solvent mask (but not a focus mask). The classification revealed 4 classes with different subunit compositions. As the occupancy of METTL15 in state 2 was low, a second round of 3D classification was performed with a focus mask around METTL15. A fraction of particles lacking METTL15 were removed from the final reconstruction to improve the map quality. The results are summarized in Fig. 6B, Supplementary Fig. 7B and Supplementary Fig. 8.

The SSU states 1 and 3 atomic models were built using the PDB 6RW5[44] as a starting model. The chain for mtIF2 was removed from state 3, while mtIF3 and mS37 were additionally removed from state 1. For SSU state 4, the PDB 7PO2[40] was used. The models were docked in the map in UCSF ChimeraX[65] and refined as two rigid bodies, one for the head and one for the body in Phenix[66]. The joints between the head and the body were regularized in Coot[67] and clashing regions were corrected manually. For SSU state 2, the PDB 7PNX[40] was docked in UCSF ChimeraX and the chain for RBFA was removed, without any further modifications to the model.

Refinement statistics for both the mt-LSU and the mt-SSU are reported in Supplementary Table 6.

## Fluorescent PhotoActivatable Ribonucleoside-enhanced Cross-Linking and ImmunoPrecipitation (fPAR-CLIP)

GTPBP8::FLAG fPAR-CLIP library preparation, sequencing and initial data processing were performed as described in ref. 45 with minor modifications. Briefly, GTPBP8::FLAG was induced with 50 ng/ml doxycycline for 48 h and next immunoprecipitated using M2 Flag beads (Merck Millipore, M8823). The unprotected RNA was digested on beads with 1 U RNase T1 (EN0541, ThermoFisher) for 15 min at RT and the beads were washed three times with RIPA buffer and three times with dephosphorylation buffer (50 mM Tris-HCl, pH 7.5, 100 mM NaCl, 10 mM MgCl₂). After washing, the protein bound RNA was dephosphorylated with Quick CIP (M0525S, New England Biolabs) for 10 min at 37 °C. Post-dephosphorylation, the beads were washed three times with dephosphorylation buffer and three times with PNK/ligation buffer (50 mM Tris-HCl, pH 7.5, 10 mM MgCl₂). Following, 0.5 μM fluorescently tagged 3' adapter (MultiplexDX) were ligated with T4 Rnl2(1–249)K227Q (M0351, New England Biolabs) overnight at 4 °C and washed three times with PNK/ligation buffer. Next, RNA footprints were phosphorylated using T4 PNK (NEB, M0201S) for 30 min in 37 °C and washed three times in RIPA buffer. To release the proteins the beads were incubated at 95 °C for 5 min in 2 × SDS Laemmli buffer and the eluates separated on 4–12% SDS/PAGE gels (NW04122BOX, Invitrogen). GTPBP8:RNA complexes were visualized on the IR 680 channel (Chemidoc MP system, Bio-Rad). Subsequently, the corresponding GTPBP8:RNA bands were excised from the gel and protein digested with Proteinase K (RPROTK-RO, Sigma Aldrich), and the released RNA isolated via phenol:chlorophorm phase separation. Following, 5' adapter ligation (MultiplexDX) was performed on the purified RNA samples with 0.5 μM of the adapter and Rnl1 T4 RNA ligase (EL0021, ThermoFisher) for 1 h at 37 °C. Next, the RNA was reverse transcribed using SuperScript IV Reverse Transcriptase (18090010, ThermoFisher) according to the manufacturer's instructions. The libraries were amplified in a series of PCR reactions performed using Platinum Taq DNA polymerase (10966034, ThermoFisher) and size selected with 3% Pippin Prep (CSD3010, Sage Science). Sequencing of the libraries was

carried out on the Illumina NovaSeq 6000 platform. For data processing Bcl2fastq (v2.20.0), Cutadapt (cutadapt 1.15 with Python 3.6.4)[68], PARpipe (https://github.com/ohlerlab/PARpipe) and Paralyzer[69] were used. The adapter sequences and primers used in the study are listed in Supplementary Table 5. Visualization of reads coverage along the mitochondrial genome was produced using IGV[70]. Standartized binding for GTPBP8 and AUH was computed by Deeptools Bigwig to matrix[71] using counts per million reads (CPM) with a bin size of 10 nucleotides. CPM ratio was plotted using R.

## Statistics
The mean values were used to generate the data from the growth curve, RT-qPCRs, and quantification analysis. The standard deviation is shown by error bars. A two-tailed, unpaired-equal variance Student's $t$-test was used for the statistical examinations. The significance cut-off was set at $p < 0.05$, with $*p < 0.05$, $**p < 0.01$, and $***p < 0.001$.

## Reporting summary
Further information on research design is available in the Nature Portfolio Reporting Summary linked to this article.

## Data availability
Ribosome profiling data are deposited at the GEO under accession number GSE242965 and at ebi under accession number S-BSST1195. The mass spectrometry proteomics data have been deposited to the ProteomeXchange Consortium via the PRIDE 46 partner repository with the dataset identifier PXD045264. Cryo-EM maps were deposited in the Electron Microscopy Data Bank (EMDB). The mt-LSU state 1 and state 2 accession codes are EMD-18460 and EMD-18461, respectively. The mt-SSU state 1, state 2, state 3, and state 4 accession codes are EMD-18438, EMD-18439, EMD-18440, and EMD-18443, respectively. Cryo-EM coordinates were deposited in the Protein Data Bank (PDB). The mt-LSU state 1 and state 2 accession codes are 8QU1, and 8QU5, respectively. The mt-SSU state 1, state 2, state 3, and state 4 accession codes are 8QRK, 8QRL, 8QRM, and 8QRN, respectively. CLIP data are deposited at the GEO under accession number: GSE256250. Source data are provided with this paper.

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

## Acknowledgements

The work was funded by Max Planck Institute, Karolinska Institute, the Knut & Alice Wallenberg Foundation (WAF2017 and KAW 2018.0080 awarded to JR, KAW 2017.0080 and 2018.0080 awarded to BMH), the Swedish Research Council (VR2016-02179, awarded to JR) and EMBO (LTF-2020-606 to MDN).

## Author contributions

Methodology: M.C., G.V.G., S.G., A.K., P.P., D.S., J.M., Y.C., M.D.N., A.A., V.L., A.A.S., J.M., X.L., I.A., J.R. Analyses: M.C., G.V.G., I.A., A.K., Y.L., M.M., M.A.P., D.B., M.H., J.R. Writing Original Draft: M.C., J.R. Review & Editing: all authors. Visualization: M.C. G.V.G. Supervision and Funding Acquisition: M.Ha., M.M., M.H., J.R.

## Funding

## Competing interests

The authors declare no competing interests.
