## [Peer Review File · Nature Communications]

GTPBP8 plays a role in mitoribosome formation in human mitochondriaREVIEWER COMMENTS

Reviewer #1 (Remarks to the Author):

This study conducted by Cipullo and colleagues unveils the cellular role of human GTPBP8 as a crucial factor in the assembly of mitochondrial ribosomes. GTPBP8 actively engages with mt-LSU during its assembly process. Notably, the absence of GTPBP8 results in diminished levels of the mitochondrial ribosome 55S monomer, thereby impairing mitochondrial translation and leading to mitochondrial dysfunction. The research data is meticulously detailed, and the logical flow of the study is coherent. While the definitive molecular function of the GTPBP8 protein remains to be fully elucidated, this study represents a significant step forward in advancing our understanding of it.

Major points:

1. The affinity purifications and BioID studies, presented in Figs 1 and 2, were executed in wild-type cells expressing tagged variants of GTPBP8 WT or mutant. The tagged versions could compete with the endogenous version of the protein for binding to interactors. Also, it raises a question as to why the authors didn't express these proteins in the GTPBP8 KO cells they had generated. Such an approach could have allowed them to assess not only the binding of tagged proteins to interactors but also whether these tagged proteins retained full functionality. Furthermore, the authors should provide insight into the levels at which these tagged proteins are overexpressed, as some mitochondrial GTPBP proteins have previously been linked to deleterious effects when present in excess.
2. Figs 2, 3, and 4 present the characterization of GTPBP8-KO cell lines. It is important to include, in all panels, the KO cell lines reconstituted with the WT version of GTPBP8, in order to eliminate potential off target effects of the gene editing approach. While these rescue cell lines are depicted in supplementary figures, it might be advisable to incorporate them into the main figures for enhanced clarity.
3. The cryo-EM structural characterization of mitoribosome particles from the KO cell line is interesting but it provides limited insights into the LSU assembly intermediates. It would be beneficial to elaborate on whether these intermediates possess other known GTPases and assembly factors, drawing further comparisons with previously reported structures of assembly intermediates. Additionally, with regards to the SSU, it might be worthwhile to comment on whether similar intermediates are observed in GTPBP5 or GTPBP10 KO cell lines for a more comprehensive explanation of their significance.

Minor points:

1. The authors should double check the protein labeling in their protein synthesis assays. Is ND4L running together with ATP8?
2. Regarding lines 430-436 and 533-536, which appear as strikethrough text, it would be necessary for the authors to clarify whether these lines are intended to be excluded from the manuscript or if they require further revision and inclusion.

Reviewer #2 (Remarks to the Author):

Cipullo et al. present a thorough in vivo analysis of the subcellular localization and function of the so far uncharacterized protein GTPBP8. The authors first show by imaging and sub-cellular fractionation that GTPBP8 localizes in mitochondria. Then, they used co-IP and proximity labeling followed by proteomics to identify potential binding partners of GTPBP8 within mitochondria. They find a large number of potential interactions, however, without a strong enrichment in mitoribosomal proteins. They next investigate the consequences of a GTPBP8 KO and find that the OxPhos system as well as mitochondrial translation in general are severely impaired. Using qPCR, they authors additionally show that GTPBP8 KO has varying effect on mitochondrial transcription, however, 12S and 16s rRNA are strongly reduced, which results in a measurable defect of ribosome formation and accumulation of early ribosome maturation states. This was then also confirmed structurally using FLAG-purified mitoribosomal subunits from GTPBP8 KO cells, which yielded cryo-EM structures of early mitoribosome assembly states of the mt-LSU as well as a range of mt-SSU states.

Although the authors present a plethora of data showing that ribosome assembly and/or maturation is impaired when GTPBP8 is missing (or its GTPase activity in particular), the underlying molecular mechanism remains somewhat remains. This is mainly due to the unfortunate lack of direct interaction of GTPBP8 with the mitoribosome itself (as evident from the co-IP and proximity labeling). However, the authors could speculate and potentially investigate additional mechanisms that might be the root cause for their observations (see points below). The presented experimental work is done well.

After the points below have been addressed by the authors, I recommend the manuscript for publication.

Christian Dienemann (signed review)

1) This comment comes with a disclaimer, that I am not an expert in mitoribosome biogenesis, however, the following came to my attention nonetheless. It is unfortunate, that GTPBP8 does not strongly interact with the mt-LSU/SSU. On one hand, this makes it difficult to structurally investigate its role in assembly/maturation of the mitoribosome. On the other hand, this lets me wonder, if GTPBP8 is even involved in ribosome maturation by directly acting on early ribosomes. In fact, some of the author's results indicate that GTPBP8 might be important for maturation of the 16S (and 12S) rRNA. For example, Figure S1 and Figure 1 show enrichment for RNA metabolism proteins like TRMT10C and RPUSD4 that act on the rRNA to introduce nucleobase modifications like pseudouridine. These modifications may have an impact on RNA stability and secondary structure. Have the authors considered incomplete RNA modification as a cause for the stalled ribosome maturation? More specifically, did the authors consider checking for alterations in rRNA modifications after GTPBP8 KO by mass spectrometry (like in <https://www.biorxiv.org/content/10.1101/2023.05.24.542018v1>)? Do the disordered regions of the 16S rRNA observed in the cryo-EM structure contain nucleobase modifications that might be crucial for rRNA folding? The finding that 12/16S rRNA levels are strongly reduced upon GTPBP8 KO while ribosomal transcription seems altered but not overall downregulated further suggests an issue with the stability of these two RNAs.

2) From the text and Figure 5 it remains unclear to me (as a non-expert), whether the mt-SSU complexes found by cryo-EM actually represent "healthy" assembly intermediates on the pathway to a mature mt-SSU. This should be made clearer. Especially, that the GTPBP8 KO affects the 12S rRNA expression level, but the effect on subunit maturation seems much less pronounced compared to the mt-LSU. What is different here?

3) p5, line 2: there is a word duplication of "in human cells"

4) The authors provide co-IP and proximity labeling results to investigate the possible interaction partners of GTPBP8. However, in the manuscript these results seem somewhat redundant. Could the authors clarify or emphasize what additionally can be learned from the having both?

5) P 10, line 6 and therein: There are instances of text that is crossed out (strikethrough)

6) Figure 5: I is a bit difficult to infer what is shown in Fig 5A as the densities are just shown in grey. Could the differences between State 1 and State 2 be highlighted by color? Additionally, as the 16S rRNA seems to be a central point here, maybe consider to color it differently as the ribosomal proteins.

7) Figure S5B and S6C, the y-axis label has to be changed. The y-axis represents the Fourier shell correlation (FSC), not the resolution.

Reviewer #3 (Remarks to the Author):

The authors characterised the protein GTPBP8. GTPBP8 was shown to be localised in the mitochondrial matrix. Flag-IP and BIO-ID mass spectrometry results show that GTPBP8 interacts with the mitochondrial expression machinery. GTPBP8 knock-out cell lines showed a defect in cellular fitness. The absence of GTPBP8 resulted in a defect in the expression of mitochondrial encoded proteins and in the assembly of OXPHOS protein complexes. GTPBP8 was also shown to be essential for mitochondrial translation and mitoribosome formation. Interestingly, disruption of the GTPBP8 G-domain had a similar effect as the loss of GTPBP8. Loss of GTPBP8 leads to accumulation of mitoribosome intermediates, suggesting that GTPBP8 plays an important role in mitoribosome assembly.

The manuscript is clearly structured and very easy to understand. The data in the figures are clearly presented and explained. The work shows an important step in understanding mitochondrial ribosome assembly and maturation and the importance of this uncharacterised protein GTPBP8 for mitochondrial function. The results support the conclusions drawn. Suppl. Tables and Method parts need to be revised (see minor points). Please include a discussion of the results obtained by mutating the G domain.

Minor points for revision

As the mass spectrometry data always contain more than is discussed and serve as a compendium to characterise GTPBP8 in more detail, I recommend that the data be presented more clearly here:

Suppl Tab 5,6,7: The MS data are confusing. Please help the reader by labelling the entries in the table header more clearly and according to the text (e.g. logFCGTPBP8_S124AvsMTS). Which log was used. How many replicates? Can the replicates still be included in this table along with additional information showing the quality of the MS data? How many peptides identified? Please make the Suppl tabs clearer in general so that readers can better understand and use these data.

Analysis of MS data

IP was analysed by MaxQuant as seen in a typical proteingroups file in PRIDE. No description could be found for MaxQuant (version, settings). Please add details here. Are IP samples also analysed in DIA mode?

PXD045264 PRIDE repository:

Sample Processing Protocol is difficult to read as several experiments are displayed together and punctuation is missing. Some special characters are also not readable. Please revise the text and give the reader more structure.

The description of data processing is not complete. Please add data analysis for IP (MaxQuant).

Methods:

MS sample preparation:

C) 100 μ l GuHCl/Tris pH 8.0, please add details,

C) How much trypsin?

Figures:

2B, please indicate number of replicates

Text:

The text contains crossed-out parts. Please correct.

REVIEWER COMMENTS

We would like to thank all referees for investing their time in evaluating our manuscript and raising pertinent questions, which, as elaborated in our responses below, have been instrumental in improving our manuscript.

We would also like to report a few alterations outlined in the results section:

A) We performed an additional crosslinking and immunoprecipitation (CLIP) assay on GTPBP8::FLAG overexpressing cells (Figure 7) to further clarify the GTPBP8 interactome network. We found that GTPBP8 is an RNA-binding protein that specifically interacts with the mitoribosomal large subunit (16S) mt-rRNA, confirming its involvement in the mtLSU assembly process. We have added a new section in the results: 'GTPBP8 is an RNA-binding protein that interacts specifically with the 16S mt' which describes this data. Additional researchers who were involved in CLIP analysis are added to the author list.

B) The inquiries of Referee 1 prompted us to perform a more thorough analysis of our mtLSU and mtSSU cryoEM data, including further focused classifications of these highly heterogenous samples. This revealed two additional mtSSU classes: i) METTL15-containing class (now State 2) and ii) mtIF2 and fMet-tRNA^{Met}-containing state (State 4). Both additional states are in agreement with the previously described final states of mtSSU assembly and translation initiation (Itoh et al., PMID: 35676484) and are in line with the recently proposed model of the mtSSU maturation (Khawaja et al., PMID:37169615). This additional analysis strongly confirms the coexistence of both mtSSU sub-assemblies and mature mtSSU particles in GTPBP8 knock-out cells, and therefore the interpretation of our data is unaffected.

C) The inquiries of Referee 3 led us to discover a mistake in the FLAG-IP Excel mass spectrometry file, which occurred during the initial data export where the highest hits' adjusted p-value was automatically converted to 0. We have rectified this mistake as shown in the updated mass spectrometry files and Figure 1A. Of note, the GTPBP interactome remains unchanged, but p-values are amended.

Reviewer #1 (Remarks to the Author):

This study conducted by Cipullo and colleagues unveils the cellular role of human GTPBP8 as a crucial factor in the assembly of mitochondrial ribosomes. GTPBP8 actively engages with mt-LSU during its assembly process. Notably, the absence of GTPBP8 results in diminished levels of the mitochondrial ribosome 55S monomer, thereby impairing mitochondrial translation and leading to mitochondrial dysfunction. The research data is meticulously detailed, and the logical flow of the study is coherent. While the definitive molecular function of the GTPBP8 protein remains to be fully elucidated, this study represents a significant step forward in advancing our understanding of it.

Major points:

1. The affinity purifications and BioID studies, presented in Figs 1 and 2, were executed in wild-type cells expressing tagged variants of GTPBP8 WT or mutant. The tagged versions could compete with the endogenous version of the protein for binding to interactors. Also, it

raises a question as to why the authors didn't express these proteins in the GTPBP8 KO cells they had generated. Such an approach could have allowed them to assess not only the binding of tagged proteins to interactors but also whether these tagged proteins retained full functionality. Furthermore, the authors should provide insight into the levels at which these tagged proteins are overexpressed, as some mitochondrial GTPBP proteins have previously been linked to deleterious effects when present in excess.

We agree with the reviewer, and we have now addressed these valid points.

The primary reason for choosing our current approach is it has proven to be robust and well-optimized. Our colleagues and collaborators have successfully utilized this robust protocol for numerous mitochondrial proteins, as evidenced by studies such as those by Jiang et al. (PMID: 31036713) and Antonicka et al. (PMID: 32877691). We chose to maintain continuity with this established method rather than introducing additional alterations.

Regarding the possible competition between the expression of the GTPBP8 tagged versions and the endogenously expressed GTPBP8, we below include a comparison of the expression levels of these proteins via Western Blot (Figure iA). The expression levels of GTPBP8-BirA* and GTPBP8^{S124A}-BirA*, under the doxycycline induction conditions used in the BioID approach, are considerably higher than the expression levels of endogenous GTPBP8. The levels of overexpression induced by doxycycline in this method are standardized at very high concentrations (1ug/ml f.c.) as this allows to increase the chances of capturing the interactions, as shown by several studies (Wanschers et al., 2012 PMID: 22238375; Rozanska et al., 2017 PMID: 28512204; Jiang et al., 2019 PMID: 31036713; Antonicka et al., 2020 PMID: 32877691).

As pointed out by the reviewer, the GTPBP8-BirA* and GTPBP8^{S124A}-BirA* constructs may interfere with GTPBP8 functionality and cause deleterious effects when expressed in excess. To address this issue, as performed for GTPBP8^{RESCUE_wt} and GTPBP8^{RESCUE_S124A}, we generated GTPBP8^{KO1} cell line re-expressing GTPBP8-BirA* (GTPBP8^{RESCUE_GTPBP8-BirA*} cell line) and GTPBP8^{S124A}-BirA* (GTPBP8^{RESCUE_GTPBP8S124A -BirA*}) (Figure iiA and Figure iiB) and performed [³⁵S]-labelling assay (Figure iiB). Similarly to GTPBP8^{RESCUE_wt} and GTPBP8^{RESCUE_S124A}, GTPBP8^{RESCUE_GTPBP8-BirA*} recovered GTPBP8^{KO1} translation defect fully when compared to HEK293 cell lines, whereas GTPBP8^{RESCUE_GTPBP8S124A -BirA*} translation levels were comparable to GTPBP8^{KO1}, GTPBP8^{KO2} and GTPBP8^{RESCUE_S124A}. With this experiment, we were able to confirm that GTPBP8-BirA* overexpression does not interfere with GTPBP8 functionality in contrast to the S124A mutated version, which, as assessed in GTPBP8^{RESCUE_S124A} cell line, causes a strong impairment in GTPBP8 physiological activity.

Figure i. A. Western Blotting to investigate GTPBP8 BioID constructs overexpression levels in comparison to GTPBP8 endogenous levels in HEK293 upon 1 ug/ml doxycycline induction. Membranes were probed with antibodies against HA-tag, GTPBP8 and GAPDH (loading control). The asterisk corresponds to an unspecific band whereas the arrow indicates the BioID constructs. B. Western Blotting to test GTPBP8^{RESCUE_GTPBP8-BirA*} and GTPBP8^{RESCUE_GTPBP8S124A-BirA*} overexpression levels under 1 ug/ml of doxycycline induction (left panel). [³⁵S]-labelling of mitochondrial translation in HEK293 control cells, GTPBP8^{RESCUE_GTPBP8-BirA*} cells and GTPBP8^{RESCUE_GTPBP8S124A-BirA*} under 1 ug/ml of doxycycline induction (right panel).

2. Figs 2, 3, and 4 present the characterization of GTPBP8-KO cell lines. It is important to include, in all panels, the KO cell lines reconstituted with the WT version of GTPBP8, in order to eliminate potential off target effects of the gene editing approach. While these rescue cell lines are depicted in supplementary figures, it might be advisable to incorporate them into the main figures for enhanced clarity.

Thank you for this suggestion. We now show these data in Figure 5. For better clarity and structure of the manuscript, we have kept the rescue characterization separated from the initial knock-out characterization.

3. The cryo-EM structural characterization of mitoribosome particles from the KO cell line is interesting but it provides limited insights into the LSU assembly intermediates. It would be beneficial to elaborate on whether these intermediates possess other known GTPases and assembly factors, drawing further comparisons with previously reported structures of assembly intermediates. Additionally, with regards to the SSU, it might be worthwhile to comment on whether similar intermediates are observed in GTPBP5 or GTPBP10 KO cell lines for a more comprehensive explanation of their significance.

After thoroughly analyzing and further processing the cryo-EM datasets, we did not find any classes of mt-LSU particles associated with mt-LSU assembly factors apart from the MALSU1:L0R8F8:mt-ACP module (State 1 and 2, Fig 6A). Nevertheless, the presence of mt-LSU assembly intermediates with highly disordered 16S mt-rRNA strongly suggests GTPBP8's involvement in earlier assembly stages compared to previously documented

structures of mt-LSU assembly intermediates (reviewed in: Khawaja et al., PMID:37169615). This is now highlighted in the discussion:

“GTPBP8 knock-out cells exhibited only mt-LSU assembly intermediates, with no fully matured mt-LSU subunits detected (Figure 6A). Notably, the mt-LSU intermediates accumulated in GTPBP8 knock-out cells are among the earliest mt-LSU intermediates identified thus far, as they display partly disordered 16S mt-rRNA domain I and II (H33-35a) and completely unstructured domain IV and V (H67-71, H89-93, H80-81) (Figure 6A). An early defect in mt-LSU assembly in GTPBP8 knock-out cells may thus be the explanation for the manifestation of more severe effects on the mt-SSU. “

Regarding the mt-SSU, there have been no studies reporting mt-SSU structures isolated from GTPBP5 or GTPBP10 knock-out cell lines thus far.

To date, only one study has been published that specifically examines the structural aspects of mt-SSU assembly in the context of dysregulated mitochondrial large subunit mt-LSU biogenesis (Itoh et al., PMID:35676484). In that report, we depleted MRM3, a mt-LSU assembly factor, which led to the accumulation of the late-stage assembly intermediates of mt-SSU (lacking mS37) as well as mature mt-SSU particles. What's particularly striking is that despite the high instability of mt-SSU in GTPBP8 KO (indicating a mechanism preventing mtSSU production/or stabilisation during its early assembly stages) the mt-SSU complexes that can be retrieved and analyzed by cryoEM resemble the mt-SSU stages found in the MRM3 KO model. This suggests an overall conservation of the late stages of mt-SSU assembly across different models.

Additional studies, including those involving GTPBP5 and GTPBP10 knock-out KO models as suggested by the reviewer, are necessary. We have included further notes in the discussion section on the potential coordination of mt-LSU and mt-SSU assembly that requires further investigations.

Minor points:

1. The authors should double check the protein labeling in their protein synthesis assays. Is ND4L running together with ATP8?

Thank you for pointing out this mistake. We have now fixed the labelling of all the [³⁵S] translation assay figures.

2. Regarding lines 430-436 and 533-536, which appear as strikethrough text, it would be necessary for the authors to clarify whether these lines are intended to be excluded from the manuscript or if they require further revision and inclusion.

The strikethrough text is not part of the manuscript and we have now removed it.

Reviewer #2 (Remarks to the Author):

Cipullo et al. present a thorough in vivo analysis of the subcellular localization and function of the so far uncharacterized protein GTPBP8. The authors first show by imaging and sub-cellular fractionation that GTPBP8 localizes in mitochondria. Then, they used co-IP and proximity labeling followed by proteomics to identify potential binding partners of GTPBP8 within mitochondria. They find a large number of potential interactions, however, without a strong enrichment in mitoribosomal proteins. They next investigate the consequences of a

GTPBP8 KO and find that the OxPhos system as well as mitochondrial translation in general are severely impaired. Using qPCR, the authors additionally show that GTPBP8 KO has varying effect on mitochondrial transcription, however, 12S and 16S rRNA are strongly reduced, which results in a measurable defect of ribosome formation and accumulation of early ribosome maturation states. This was then also confirmed structurally using FLAG-purified mitoribosomal subunits from GTPBP8 KO cells, which yielded cryo-EM structures of early mitoribosome assembly states of the mt-LSU as well as a range of mt-SSU states. Although the authors present a plethora of data showing that ribosome assembly and/or maturation is impaired when GTPBP8 is missing (or its GTPase activity in particular), the underlying molecular mechanism somewhat remains. This is mainly due to the unfortunate lack of direct interaction of GTPBP8 with the mitoribosome itself (as evident from the co-IP and proximity labeling). However, the authors could speculate and potentially investigate additional mechanisms that might be the root cause for their observations (see points below). The presented experimental work is done well.

After the points below have been addressed by the authors, I recommend the manuscript for publication.

Christian Dienemann (signed review)

1) This comment comes with a disclaimer, that I am not an expert in mitoribosome biogenesis, however, the following came to my attention nonetheless. It is unfortunate, that GTPBP8 does not strongly interact with the mt-LSU/SSU. On one hand, this makes it difficult to structurally investigate its role in assembly/maturation of the mitoribosome. On the other hand, this lets me wonder, if GTPBP8 is even involved in ribosome maturation by directly acting on early ribosomes. In fact, some of the author's results indicate that GTPBP8 might be important for maturation of the 16S (and 12S) rRNA. For example, Figure S1 and Figure 1 show enrichment for RNA metabolism proteins like TRMT10C and RPUSD4 that act on the rRNA to introduce nucleobase modifications like pseudouridine. These modifications may have an impact on RNA stability and secondary structure. Have the authors considered incomplete RNA modification as a cause for the stalled ribosome maturation? More specifically, did the authors consider checking for alterations in rRNA modifications after GTPBP8 KO by mass spectrometry (like in <https://www.biorxiv.org/content/10.1101/2023.05.24.542018v1>)? Do the disordered regions of the 16S rRNA observed in the cryo-EM structure contain nucleobase modifications that might be crucial for rRNA folding? The finding that 12/16S rRNA levels are strongly reduced upon GTPBP8 KO while ribosomal transcription seems altered but not overall downregulated further suggests an issue with the stability of these two RNAs.

We thank the reviewer for the important comment.

Initially, we were hoping to identify potential alterations in the modification status of mitoribosomes in the GTPBP8 knockout cells using cryoEM approach. However, as pointed out by the Reviewer, all essential methylation and pseudouridylation sites of mt-LSU reside within the region of 16S rRNA which remains highly unstructured upon GTPBP8 depletion. We have now highlighted in Figure 6A all known modification sites of 16S. (Of note, the methylation introduced by MRM1 is non-essential and it occurs at a frequency of approximately 50% (Singh et al., <https://doi.org/10.1101/2023.05.24.542018>).

After consulting with proteomics experts, we concluded that conducting mass spectrometry analysis to assess rRNA modification status in GTPBP8 knockout cells would likely not be feasible. This is primarily due to the significant decrease in rRNA levels observed in GTPBP8 KO (less than 20% of the wild-type 16S), posing a technical challenge as large quantities of rRNA are necessary for this experiment (for mass spectrometry analysis of the wild-type

mitoribosomes, 8 liters of cultured cells were previously used, *personal communication with V Singh*, Singh et al., <https://doi.org/10.1101/2023.05.24.542018>).

Instead, we decided to perform a CLIP assay to identify potential GTPBP8-rRNA interactions. We have now included data from this assay in the result section 'GTPBP8 is an RNA-binding protein that interacts specifically with the 16S mt-rRNA'.

We found that GTPBP8 binds to the 16S mt-rRNA (Figure 7), confirming our conclusions drawn from cryoEM analyses. This interaction is highly specific to 16S rRNA only.

Interestingly, although the coverage of CLIP reads is spread across the 16S mt-rRNA, we could identify several pronounced peaks (Figure iiA). However, none of the identified peaks is in the vicinities of 16S mt-rRNA modifications.

This is so far the only report applying CLIP assay to study protein-mtLSU interactions, therefore, unfortunately, we cannot perform a comparative analysis of binding sites with other mt-LSU assembly factors to exclude technical biases of the assay. This prompts us to exclude a detailed description of the locations of the peaks in the manuscript.

Based on our data, while we cannot dismiss the possibility of GTPBP8 involvement in rRNA modification, further comprehensive analyses are required to elucidate its exact role in 16S mt-rRNA maturation. This is further highlighted in the discussion section.

Figure ii. A. Alignment of GTPBP8::FLAG CLIP reads to the mtDNA region encompassing 16S rRNA (MT-RNR2). The numbers correspond to the highest peaks on 16S mt-rRNA. C to T conversions introduced by CLIP are indicated in red.

2) From the text and Figure 5 it remains unclear to me (as a non-expert), whether the mt-SSU complexes found by cryo-EM actually represent “healthy” assembly intermediates on the pathway to a mature mt-SSU. This should be made clearer. Especially, that the GTPBP8 KO affects the 12S rRNA expression level, but the effect on subunit maturation seems much less pronounced compared to the mt-LSU. What is different here?

Following the biochemical characterisation of GTPBP8 knock-out, our initial assumption was that GTPBP8 is indeed involved in the maturation of both subunits. However, the cryoEM data clearly indicate a more pronounced impact of GTPBP8 KO on the mtLSU biogenesis, compared to the mtSSU biogenesis, as mentioned by the reviewer. This observation is now further supported by our CLIP data which indicate GTPBP8 specific binding to the 16S mt-rRNA and not to the 12S mt-rRNA.

The presence of 'healthy' late-stage assembly intermediates of mtSSU and mature mtSSU can be explained by GTPBP8's indirect influence on mtSSU biogenesis. It is tempting to speculate that crosstalk between the assembly of mtLSU and mtSSU exists at different stages of these processes. This is highlighted now in the discussion section. As suggested by the reviewer, we also highlighted the fact that mtSSU complexes produced in GTPBP8 KO are fully matured and ready to engage in translation.

3) p5, line 2: there is a word duplication of "in human cells"

Thank you for spotting this mistake. It is now fixed.

4) The authors provide co-IP and proximity labeling results to investigate the possible interaction partners of GTPBP8. However, in the manuscript these results seem somewhat redundant. Could the authors clarify or emphasize what additionally can be learned from the having both?

To study GTPBP8 interactome, we initially performed the FLAG-IP experiment as described in our previous works on mitoribosome-associated GTPases (Cipullo et al., PMID: 33283228; Busch et al., PMID: 31693908). However, as this method did not reveal robust binding partners for GTPBP8, we opted for an alternative approach, BioID, to explore potential more transient interactions. Both methodologies did not pinpoint specific interactors but they both suggested that GTPBP8 is involved in mitoribosome biogenesis. Rather than redundant, we believe that this observation, confirmed with two different methods, strengthens our initial hypothesis and therefore we would like to keep it in the manuscript.

5) P 10, line 6 and therein: There are instances of text that is crossed out (strikethrough)

We have now deleted all the crossed-out instances.

6) Figure 5: I is a bit difficult to infer what is shown in Fig 5A as the densities are just shown in grey. Could the differences between State 1 and State 2 be highlighted by color? Additionally, as the 16S rRNA seems to be a central point here, maybe consider to color it differently as the ribosomal proteins.

Thank you for this useful comment. We have modified Figure 6A (previous 5A) accordingly by showing the 16S mt-rRNA in light blue color and by highlighting the 16S mt-rRNA disordered helices via a comparison with the mature mtLSU (yellow).

7) Figure S5B and S6C, the y-axis label has to be changed. The y-axis represents the

Fourier shell correlation (FSC), not the resolution.

We have now corrected the y-axis label for both Figure S5B and S6C.

Reviewer #3 (Remarks to the Author):

The authors characterised the protein GTPBP8. GTPBP8 was shown to be localised in the mitochondrial matrix. Flag-IP and BIO-ID mass spectrometry results show that GTPBP8 interacts with the mitochondrial expression machinery. GTPBP8 knock-out cell lines showed a defect in cellular fitness. The absence of GTPBP8 resulted in a defect in the expression of mitochondrial encoded proteins and in the assembly of OXPHOS protein complexes. GTPBP8 was also shown to be essential for mitochondrial translation and mitoribosome formation. Interestingly, disruption of the GTPBP8 G-domain had a similar effect as the loss of GTPBP8. Loss of GTPBP8 leads to accumulation of mitoribosome intermediates, suggesting that GTPBP8 plays an important role in mitoribosome assembly. The manuscript is clearly structured and very easy to understand. The data in the figures are clearly presented and explained. The work shows an important step in understanding mitochondrial ribosome assembly and maturation and the importance of this uncharacterised protein GTPBP8 for mitochondrial function. The results support the conclusions drawn. Suppl. Tables and Method parts need to be revised (see minor points). Please include a discussion of the results obtained by mutating the G domain.

Thank you for the positive feedback and constructive input. We now discussed in more detail the results obtained by the mutated G-domain GTPBP8 protein.

“Of note, no difference in the enrichment analysis was observed between the pull-downs performed with wild-type GTPBP8-BirA* and its mutated version (GTPBP8S124A-BirA*). However, complementation experiments confirmed the functionality of the overexpressed GTPBP8 but not GTPBP8S124A (Figure 5). These results suggest that the S124A mutation interferes with GTPBP8 hydrolysis but not with GTPBP8 binding, which may occur when GTPBP8 is in the GTP-bound state.”

Moreover, as suggested by Reviewer 1, we moved data from Figure S4 B-E which contains complementation experiments performed with a GTPBP8 variant with a mutation in the G-domain (S124A) to the main Figure 5, to highlight the functional importance of the GTPase activity of GTPBP8.

Minor points for revision

As the mass spectrometry data always contain more than is discussed and serve as a compendium to characterise GTPBP8 in more detail, I recommend that the data be presented more clearly here:

Suppl Tab 5,6,7: The MS data are confusing. Please help the reader by labelling the entries in the table header more clearly and according to the text (e.g. logFCGTPBP8_S124AvsMTS).

Column names were renamed in accordance with the text and additional information was provided about the content of each column in the updated supplementary tables (Table S6 for

the FLAG-IP experiment, Table S7 for the mitoproteome experiment and Table S8 for the BioID experiment).

Which log was used.

Differences in abundance are reported on a log₂ scale. We included this information when missing in the main text and supplementary tables.

How many replicates? Can the replicates still be included in this table along with additional information showing the quality of the MS data? How many peptides identified?

We used three or four replicates in each of the proteomics experiments. We expanded our supplementary tables information by including the protein intensity values in each sample, the numbers of quantified peptides and proteins and the correlation coefficients between the protein intensity values in each sample. All this information can be found in each table file (Table S6 for the FLAG-IP experiment, Table S7 for the mitoproteome experiment and Table S8 for the BioID experiment).

Please make the Suppl tabs clearer in general so that readers can better understand and use these data.

We have now modified the supplementary tables' tabs for the reader to better understand the data.

Analysis of MS data

IP was analysed by MaxQuant as seen in a typical protein groups file in PRIDE. No description could be found for MaxQuant (version, settings). Please add details here. Are IP samples also analysed in DIA mode?

We thank the reviewer for pointing out this omission. The IP samples were analyzed using data dependent acquisition and not DIA. We included a description of the mass spectrometry parameters for the analysis of the IP samples as well as the computational analysis in the "Material and Methods" section as follows:

"IP samples were analyzed using data-dependent acquisition (DDA). Peptides were separated on a 25 cm, 75 µm internal diameter PicoFrit analytical column (New Objective) packed with 1.9 µm ReproSil-Pur 120 C18-AQ media (Dr. Maisch,) using an EASY-nLC 1200 (Thermo Fisher Scientific). The column was maintained at 50°C. Buffer A and B were 0.1% formic acid in water and 0.1% formic acid in 80% acetonitrile. Peptides were separated on a segmented gradient from 6% to 31% buffer B for 45 min and from 31% to 50% buffer B for 5 min at 200 nl / min. Eluting peptides were analyzed on QExactive HF mass spectrometer (Thermo Fisher Scientific). Peptide precursor m/z measurements were carried out at 60000 resolution in the 300 to 1800 m/z range. The ten most intense precursors with charge state from 2 to 7 only were selected for HCD fragmentation using 25% normalized collision energy. The m/z values of the peptide fragments were measured at a resolution of 30000 using a minimum AGC target

of 2e5 and 80 ms maximum injection time. Upon fragmentation, precursors were put on a dynamic exclusion list for 45 sec.”

PXD045264

PRIDE

repository:

Sample Processing Protocol is difficult to read as several experiments are displayed together and punctuation is missing. Some special characters are also not readable. Please revise the text and give the reader more structure. The description of data processing is not complete. Please add data analysis for IP (MaxQuant).

We updated the text to improve its structure. The information about the mass spectrometric analysis of the IP samples and the computational analysis using MaxQuant is also included in the “Material and Methods” section as follows:

“GTPBP8 IP raw data were analyzed with MaxQuant version 1.6.1.0⁵⁰ using the integrated Andromeda search engine⁵¹. Peptide fragmentation spectra were searched against the canonical sequences of the human reference proteome (proteome ID UP000005640, downloaded September 2018 from UniProt). Methionine oxidation and protein N-terminal acetylation were set as variable modifications; cysteine carbamidomethylation was set as fixed modification. The digestion parameters were set to “specific” and “Trypsin/P,” The minimum number of peptides and razor peptides for protein identification was 1; the minimum number of unique peptides was 0. Protein identification was performed at a peptide spectrum matches and protein false discovery rate of 0.01. The “second peptide” option was on. Successful identifications were transferred between the different raw files using the “Match between runs” option. Label-free quantification (LFQ)⁵² was performed using an LFQ minimum ratio count of two.”

Methods:

MS sample preparation:

C) 100 µl GuHCl/Tris pH 8.0, please add details

We have included more details in the “Material and Methods” section as follows:

“Mitochondrial pellets (50 µg) were resuspended in 100 µl of GuHCl/Tris pH 8.0 solution (6M GuHCl solution in 100 mM Tris-HCl pH 8.0) and sonicated twice for 5 minutes (10 seconds ON/OFF cycles).”

C) How much trypsin?

We have now provided the information that trypsin was added at a 1:50 w/w ratio in the “Material and Methods” section.

Figures:

2B, please indicate number of replicates

Thank you for pointing out this omission. We have now reported the number of replicates in Figure 2B.

Text:

The text contains crossed-out parts. Please correct.

Thank you for spotting this, we have now deleted all the crossed-out parts.

REVIEWERS' COMMENTS

Reviewer #1 (Remarks to the Author):

The authors have satisfactorily responded to all my previous queries and those by the other reviewers. Therefore, I recommend the acceptance of this manuscript for publication.

Reviewer #2 (Remarks to the Author):

The authors addressed all my points and present an overall much improved manuscript.

Reviewer #3 (Remarks to the Author):

A new fPAR-CLIP experiment significantly improved the manuscript. It showed that GTPBP8 is an RNA-binding protein that interacts with the 16S rRNA large subunit.

My points regarding the MS data were all addressed in the revised manuscript. The authors described the improvements made to the additional tables, but I could not find the information in the data provided (maybe I didn't find the link). However, since the concerns have been addressed and the improvements sound satisfactory, I recommend accepting the manuscript.